# Water scarcity hotspots travel downstream due to human interventions in the 20th and 21st century

T.I.E. Veldkamp[1,2], Y. Wada[2,3,4,5], J.C.J.H. Aerts[1,6], P. Döll[7,8], S.N. Gosling[9], J. Liu[10], Y. Masaki[11,12], T. Oki[13], S. Ostberg[14,15], Y. Pokhrel[16], Y. Satoh[2], H. Kim[13] & P.J. Ward[1]

Water scarcity is rapidly increasing in many regions. In a novel, multi-model assessment, we examine how human interventions (HI: land use and land cover change, man-made reservoirs and human water use) affected monthly river water availability and water scarcity over the period 1971–2010. Here we show that HI drastically change the critical dimensions of water scarcity, aggravating water scarcity for 8.8% (7.4–16.5%) of the global population but alleviating it for another 8.3% (6.4–15.8%). Positive impacts of HI mostly occur upstream, whereas HI aggravate water scarcity downstream; HI cause water scarcity to travel downstream. Attribution of water scarcity changes to HI components is complex and varies among the hydrological models. Seasonal variation in impacts and dominant HI components is also substantial. A thorough consideration of the spatially and temporally varying interactions among HI components and of uncertainties is therefore crucial for the success of water scarcity adaptation by HI.

[1] Institute for Environmental Studies (IVM), Vrije Universiteit Amsterdam, De Boelelaan 1085, Amsterdam 1081 HV, The Netherlands. [2] International Institute for Applied Systems Analysis, Laxenburg A-2361, Austria. [3] Center for Climate Systems Research, Columbia University, New York, New York 10027, USA. [4] NASA Goddard Institute for Space Studies, New York, New York 10027, USA. [5] Department of Physical Geography, Utrecht University, Utrecht 3508 TC, The Netherlands. [6] Department of Geography, University of California–Santa Barbara, Santa Barbara 93106, California, USA. [7] Institute of Physical Geography, Goethe University Frankfurt, Frankfurt am Main 60438, Germany. [8] Senckenberg Biodiversity and Climate Research Center (BiK-F), Frankfurt am Main 60325, Germany. [9] School of Geography, University of Nottingham, Nottingham NG7 2RD, UK. [10] School of Environmental Science and Engineering, South University of Science and Technology of China, Guangdon 518055, China. [11] National Institute for Environmental Studies, Tsukuba 3058506, Japan. [12] Hirosaki University, Hirosaki 0368561, Japan. [13] Institute of Industrial Science, The University of Tokyo, Tokyo 153-8505, Japan. [14] Potsdam Institute for Climate Impact Research, Potsdam 14473, Germany. [15] Geography Department, Humboldt-Universität zu Berlin, D-10099 Berlin, Germany. [16] Department of Civil and Environmental Engineering, Michigan State University, East Lansing, Michigan 48824-2604, USA. Correspondence and requests for materials should be addressed to T.I.E.V. (email: ted.veldkamp@vu.nl).

Socioeconomic developments increasingly put pressure on our global freshwater resources, thereby increasing water scarcity, that is, the temporal deficits in freshwater resources compared with anthropogenic and environmental demands[1]. Over the past 100 years, human water demand increased almost 8-fold[2] due to the quadrupling of the global population, increases in per capita food demands and rising standards of living[3–7]. Increasing volumes of water are needed to feed the global population and to drive local and global economies[4]. To keep up with these growing demands, large-scale human interventions (HI) have taken place: land use and land cover change (LULCC), including irrigation to increase food productivity; dams and reservoirs to control the timing of streamflow; and water withdrawals from surface water bodies and groundwater to fulfil water demands. Although these HI are targeted at local to regional scales, they are known to impact the hydrological cycle and can affect streamflow on larger scales, such as in downstream areas[7–17]. For example, earlier work has shown that land use conversions from forest to cropland increase river discharge, whereas irrigation increases local runoff but decreases basin-wide runoff due to increased evapotranspiration rates[9,15]. Moreover, upstream water withdrawals from the streamflow may decrease water availability for downstream use[9,14,17], and dams and reservoirs can severely alter the timing of low- and high-flows[8–12,17]. Having insight in the impacts of HI can help water managers to highlight locations where HI have been beneficial from a water resources perspective. Moreover, it can help to identify regions where there is still room for expansion and/or intensification of HI, given the historic and future increases in water and food demand, and the expected changes in climate conditions[18–28]. Such insights also allow us to identify regions where a more optimal incorporation of HI is required in the socio-hydrological system.

The allocation of shared water resources between upstream and downstream regions has a prominent place on the global science-policy agenda, given the fact that transboundary rivers and lakes cover almost half of the global land area and are home to ∼40% of the global population[29]. Sharing water resources creates interdependencies that may lead to cooperative or conflictive events, although evidence of causality is limited[30]. The development of event databases such as the Transboundary Fresh Water Dispute Database[31], the Water Conflict Chronology[32] and the ICOW River Claims Data Set[33] enable a more systematic assessment of water conflict and resolution processes, and provide insights into the effectiveness of cooperative arrangements[34–40]. Despite the widespread recognition of the importance of upstream–downstream interactions within river basins, only limited quantitative research has been performed to unravel and understand the dominant drivers of change, linking different types of HI with water scarcity and (increased) exposure to water scarcity[14]. At the same time, unequal impacts of upstream–downstream interactions on water resources and water scarcity at a higher spatial resolution, for example, within administrative regions or river basins, are often left unstudied, despite their potential impact on societies.

Going beyond the first assessment[14] of the impact of upstream water use on downstream water scarcity at a yearly scale, this study incorporates different types of HI, namely LULCC, dam and reservoir operations, and upstream water consumption, and compares their impacts on freshwater availability and water scarcity to the trends in climate change impacts. The central aims of this study are therefore to quantify how HI have altered the critical dimensions of water scarcity, including the average duration, occurrence and severity of water scarcity; to evaluate whether, and to what extent, HI have led to a reshuffling of water scarcity hotspots, leading to changes in the exposure to water

scarcity events; and to assess how HI over time contribute to or dominate over the trend in climate change impacts.

To do this, we performed a scenario analysis using monthly water resources simulations at a $0.5° × 0.5°$ ($\sim 50 × 50$ km at the equator) spatial resolution for the period 1971–2010 reflecting conditions of no HI (NHI) and time-varying HI (see Methods). This study uses an ensemble of five state-of-the-art global hydrological models (GHMs) allowing for robust estimates: H08 (refs 41,42), LPJmL[43,44], MATSIRO[45], PCR-GLOBWB[46,47] and WaterGAP[13]. Each of the GHMs was driven by three global state-of-the art observation-based historical climate datasets: PGFv2 (ref. 48), GSWP3 (http://hydro.iis.u-tokyo.ac.jp/GSWP3) and WFD/WFDEI[49]. The GHMs were also forced by a set of socio-economic parameters to model historical demands: gross domestic product (GDP), population density, livestock density, land use and land cover[2]. In this study, we used the HYDE 3—MIRCA data set[50–52], ensembled following Fader et al.[53], for simulating the changes in irrigation and/or cropland patterns over time and their time-varying impacts on water availability and water scarcity. We introduced a spatially and temporally explicit measure of the minimum environmental flow requirement[54–57], that is, a rough global estimate of water that ecosystems need to sustain healthy conditions. By combining this with our seasonal assessment of water availability and water scarcity, accounting for seasonal variability and regional variation at a high spatial resolution, we were able to develop an updated Water Scarcity Index (WSI)[1]. This index provides a more meaningful indicator for water scarcity at the seasonal scale than those used in past studies, as it reflects both the human and environmental water needs[58,59]. In doing so, we build upon the latest insights from previous hydrologic research[8–13,15] and translate its implications into the domain of water scarcity and exposure to water scarcity events.

Our results show that HI substantially changed the critical dimensions of water scarcity between 1971 and 2010, reshuffling hotspots of water scarcity and causing a distinct pattern of beneficiaries and losers, involving more than one-third of the global population. Large differences between upstream and downstream regions exist under the limited net global impact of HI, causing water scarcity to travel downstream. Attribution of water scarcity changes to HI components is complex and varies among the hydrological models. Seasonal variation in impacts and dominant HI components is also substantial. A systematic deliberation of the spatially and temporally varying interactions among HI components and of uncertainties is therefore needed when adapting to water scarcity by HI.

## Results

**Impacts of HI at the global scale**. HI employed to maximize the utilization of water resources have significantly changed the local water availability over the period 1971–2010 (Fig. 1 and Table 1). Although on average 20.4% (16.6–29.1%) of the global population (2010 values) experienced a significant increase in water availability due to the implementation of HI, 23.7% (18.6–39.0%) experienced a significant decrease. Consequently, HI significantly altered the critical dimensions of water scarcity (average duration, occurrence and severity) and caused a substantial reshuffling of those exposed to water scarcity, affecting a considerable share of the global population. One-third of the global population (2010) experienced a significant increase in the average duration (20.1–42.7% of the global population) and occurrence (14.4–50.4%) of water scarcity events, respectively, whereas significant decreases in average duration and occurrence due to HI were felt by 24.8% (19.0–26.4%) and 20.7% (19.4–26.8%). Whereas HI alleviated water scarcity conditions, on average, for 8.3% (6.4–15.8%) of the global population and caused 2.9% (2.2–4.5%) to move out of water scarcity, it

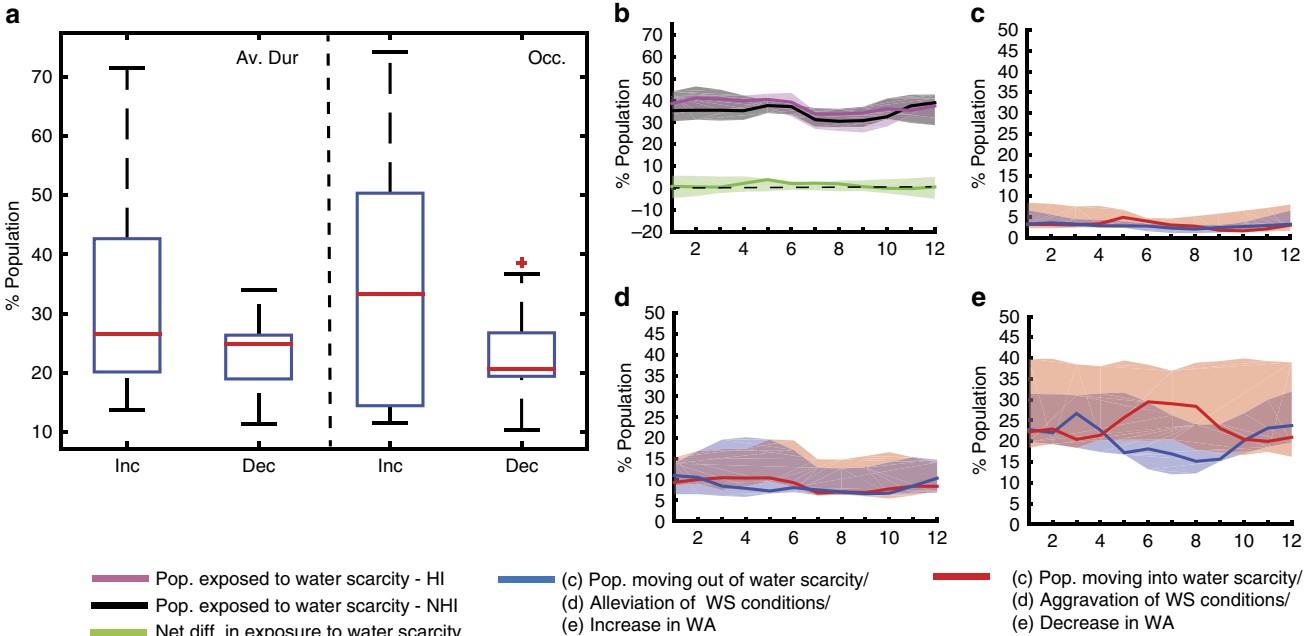

**Figure 1 | Impact of HI on water availability and water scarcity at the global scale.** (**a**) The percentage of the global population that experiences a significant increase or decrease (>5%) in average duration and occurrence (number of months) of water scarcity. (**b**) The population exposed to water scarcity and the net change in exposure due to HI compared with the NHI run (NHI), whereas (**c**) gives the population moving into/out of water scarcity and (**d**) presents the population experiencing aggravated/alleviated water scarcity (WS) conditions. (**e**) Visualizes, finally, the percentage of population experiencing increases/decreases in water availability (WA). The boxes in (**a**) and the shaded areas in (**b–e**) represent the interquartile ranges (q25–q75) and the lines the ensemble-median values. Regional values for a selection of river basins are shown in the Supplementary Material (Supplementary Figs 1–5).

resulted in aggravated water scarcity conditions for 8.8% (7.4–16.5%) and drove 3.0% (2.3–6.7%) of the population into water scarcity. As a net result, HI slightly increased (+1.2%) the global exposure of population to water scarcity. Using estimates of the relative location of impacted regions within the river basin, we find that the positive impacts of HI mostly occur in upstream areas, whereas areas located further downstream are more often impacted negatively (Fig. 2 and Table 2). For example, those populations who moved out of water scarcity due to HI have, on average, a relatively lower fraction of upstream area (0.16–0.28) compared with those who moved into water scarcity (0.23–0.35). As a result, HI caused water scarcity globally to travel downward through river basins (Fig. 2b).

**Impacts of HI at the regional scale.** Although the absolute numbers vary, we find the same overall patterns for most of the river basins studied in detail. Although the net impact of HI on the critical dimensions of water scarcity is often close to zero (Supplementary Fig. 1), populations of most basins either experienced increasing or decreasing water availability due to implementation of HI (Supplementary Fig. 2) and a substantial share of the population moved in or out of water scarcity (Supplementary Fig. 3). Moreover, a significant share of the population experienced a change in the average duration, occurrence and severity of water scarcity due to HI (Supplementary Figs 4 and 5). The mutual differences across the different basins with respect to the share of population being exposed to (i) an alleviation/aggravation of water scarcity conditions, (ii) movement in/out of water scarcity or (iii) significant increases/decreases in water availability can partly be clarified by the initial pressure on the available water resources under the NHI conditions. For example, a relative high share of the population in the Ganges–Brahmaputra and the Indus basin lived already in (deep) water scarce conditions and implementation of

HI led predominantly to an aggravation/alleviation of these conditions. In contrast, we find that a relatively large share of the population living in the Huang He basin moved into water scarcity due to HI, whereas significant aggravations of water scarcity conditions of those living already in water scarcity conditions were experienced to a lesser extent. For the Amazonas and Congo basin, we find only limited impacts of HI on the availability of water resources, with even lower effects on the critical dimensions of water scarcity. This can be explained by the relatively low rate of HI in these basins, combined with a relatively low pressure on the available water resources in major parts of these basins. Similar to the global results, movements into water scarcity due to HI are predominantly found in more downstream regions, whereas people moving out tend to live in relatively upstream areas (Supplementary Fig. 6). This regional divide can be seen in most river basins, apart from the Paraná and the Volga, where those moving into water scarcity live relatively more upstream. An interesting pattern that is hidden in the globally aggregated numbers is that exposure to water scarcity in the majority of the river basins studied shows a distinct seasonal pattern. For example, our results show a clear seasonal pattern with relatively high exposure to water scarcity in Asia (for example, Ganges–Brahmaputra and Huang He) during the northern Hemisphere spring, before the onset of the rainy season, and high exposure to water scarcity in European and American basins (for example, Mississippi, Rhine and Paraná) during the northern Hemisphere summer and autumn (Supplementary Fig. 2). The impacts of HI follow this seasonal pattern in most river basins, with highest impacts in those months with the highest pressure on the available water resources.

**Dominant drivers of change at the global and regional scale.** Changes in the availability of water resources are driven by multiple mechanisms, of which its dominance varies across basins

**Table 1 | Percentage of population exposed to significant changes in the critical dimensions of water scarcity due to HI.**

|  | Ensemble-median (q25–q75) | H08 (q25–q75) | LPJmL (q25–q75) | MATSIRO (q25–q75) | PCR-GLOBWB (q25–q75) | WaterGAP (q25–q75) |
|---|---|---|---|---|---|---|
| *Exposure to water scarcity* | | | | | | |
| Long-term mean global exposure to water scarcity under HI run | 37.6 (32.0–40.6) | 41.1 (40.3–41.6) | 35.0 (34.3–36.2) | 43.5 (42.9–44.0) | 27.1 (26.3–27.7) | 35.9 (35.4–36.5) |
| Net change in long-term mean global exposure to water scarcity due to HI | 1.2 (−2.5 to 4.4) | −2.5 (−2.8 to −2.3) | −2.7 (−2.9 to −2.4) | 6.3 (5.8 to 7.7) | 1.3 (1.2 to 1.5) | 4.2 (4.0 to 4.3) |
| Movement into water scarcity due to HI | 3.0 (2.3–6.7) | 1.9 (1.8–1.9) | 2.3 (2.2–2.5) | 9.1 (8.8–10.2) | 3.0 (2.9–3.2) | 6.6 (6.5–6.7) |
| Movement out of water scarcity due to HI | 2.9 (2.2–4.5) | 4.3 (4.2–4.6) | 4.9 (4.7–5.0) | 2.3 (2.3–2.4) | 1.8 (1.7–1.8) | 2.7 (2.6–2.7) |
| *Severity of water scarcity* | | | | | | |
| Significant aggravation of water scarcity conditions due to HI | 8.8 (7.4–16.5) | 8.5 (8.3–8.7) | 5.6 (5.5–6.0) | 22.6 (21.5–23.2) | 7.7 (7.3–8.0) | 16.1 (15.8–16.5) |
| Significant alleviation of water scarcity conditions due to HI | 8.3 (6.4–15.8) | 15.7 (15.1–16.2) | 17.6 (17.4–18.4) | 6.5 (6.3–6.6) | 8.0 (7.8–8.1) | 6.5 (6.4–6.6) |
| *Water availability* | | | | | | |
| Significant increase in water availability due to HI | 20.4 (16.6–29.1) | 28.3 (27.8–29.2) | 31.9 (31.6–32.4) | 14.4 (14.2–14.6) | 19.3 (19.2–19.7) | 17.8 (17.6–18.1) |
| Significant decrease in water availability due to HI | 23.7 (18.6–39.0) | 18.3 (18.1–18.5) | 17.6 (17.5–18.4) | 52.1 (51.4–52.7) | 23.5 (22.6–23.9) | 38.6 (38.2–39.0) |
| *Average duration and occurrence of water scarcity events* | | | | | | |
| Significant increase in average duration of water scarcity events due to HI | 26.5 (20.1–42.7) | 13.9 (13.8–14.1) | 20.9 (20.1–21.3) | 54.0 (50.8–67.1) | 26.5 (26.0–27.0) | 41.2 (40.8–42.7) |
| Significant decrease in average duration of water scarcity events due to HI | 24.8 (19.0–26.4) | 33.2 (32.1–33.8) | 25.9 (25.1–26.4) | 16.7 (12.7–17.1) | 19.4 (19.0–20.2) | 24.9 (24.7–25.2) |
| Significant increase in occurrence of water scarcity events due to HI | 33.3 (14.4–50.4) | 14.6 (14.4–15.5) | 12.0 (11.6–12.3) | 57.0 (56.0–69.9) | 33.3 (33.1–33.4) | 50.1 (49.1–50.4) |
| Significant decrease in occurrence of water scarcity events due to HI | 20.7 (19.4–26.8) | 25.5 (25.0–26.8) | 36.6 (36.3–38.1) | 15.0 (11.5–15.0) | 19.7 (19.4–20.1) | 20.7 (20.7–21.2) |

HI, human intervention.
Table 1 shows the global long-term mean share of the global population exposed to water scarcity under the HI run; the net change in exposure to water scarcity (% of the global population) due to HI; the long-term mean share of the global population that moves into/out of water scarcity due to HI; the long-term mean share of the global population that already lives in water scarcity and experiences a significant aggravation/alleviation of its water scarcity conditions due to HI; the long-term mean share of the global population that experiences significant increases/decreases in water availability due to HI; and the long-term mean share of the global population that experiences significant increases/decreases in average duration and occurrence of water scarcity due to HI. The results presented in the table show the ensemble medians per global hydrological model or over the full ensemble, with the interquartile ranges (q25,q75) between brackets.

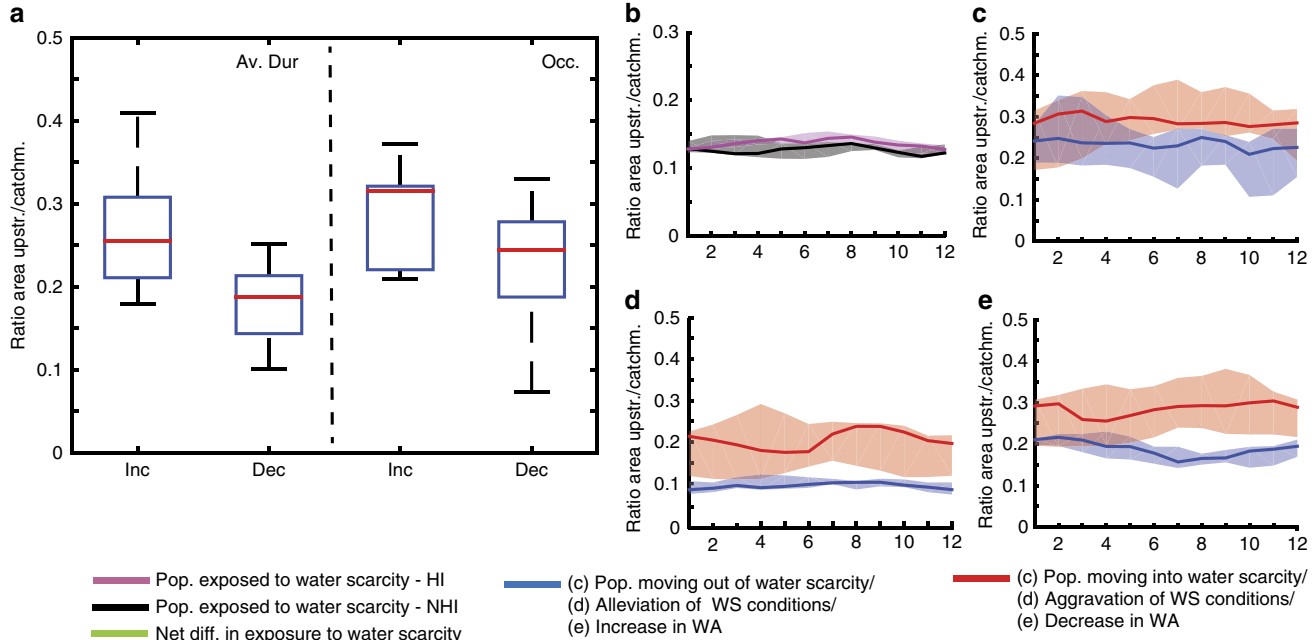

**Figure 2 | Global population-weighted mean ratio between the upstream area and total catchment area of impacted regions.** (**a**) The difference in the global population-weighted mean location within the river basin between those areas exposed to a significant increase and decrease in average duration and total occurrence of water scarcity. Higher values indicate here areas being located more downstream. (**b**) The difference in these ratios between those areas exposed to water scarcity under the NHI run (NHI) and the HI run. (**c–e**) The difference in these ratios between those areas exposed to a movement in/out of water scarcity due to HI, a significant alleviation/aggravation of water scarcity (WS) conditions and a significant increase/decrease in water availability (WA), respectively. The boxes in (**a**) and the shaded areas in (**b–e**) represent the interquartile ranges (q25–q75) and the lines the ensemble-median values.

**Table 2 | Relative location within the river basin of those being exposed to significant changes in the critical dimensions of water scarcity due to HI.**

|  | Ensemble-median (q25–q75) | H08 (q25–q75) | LPJmL (q25–q75) | MATSIRO (q25–q75) | PCR-GLOBWB (q25–q75) | WaterGAP (q25–q75) |
|---|---|---|---|---|---|---|
| *Exposure to water scarcity* | | | | | | |
| Movement into water scarcity due to HI | 0.29 (0.23–0.35) | 0.36 (0.35–0.37) | 0.37 (0.36–0.38) | 0.22 (0.22–0.23) | 0.30 (0.29–0.30) | 0.23 (0.22–0.23) |
| Movement out of water scarcity due to HI | 0.23 (0.16–0.28) | 0.33 (0.33–0.34) | 0.28 (0.27–0.29) | 0.24 (0.22–0.24) | 0.07 (0.07–0.07) | 017 (0.16–0.18) |
| *Severity of water scarcity* | | | | | | |
| Significant aggravation of water scarcity conditions due to HI | 0.21 (0.12–0.25) | 0.25 (0.25–0.26) | 0.25 (0.24–0.26) | 0.11 (0.11–0.11) | 0.21 (0.20–0.21) | 0.13 (0.12–0.13) |
| Significant alleviation of water scarcity conditions due to HI | 0.09 (0.09–0.11) | 0.09 (0.09–0.09) | 0.10 (0.10–0.10) | 0.11 (0.10–0.12) | 0.07 (0.07–0.07) | 0.11 (0.11–0.12) |
| *Water availability* | | | | | | |
| Significant increase in water availability due to HI | 0.19 (0.16–0.21) | 0.20 (0.20–0.20) | 0.19 (0.18–0.19) | 0.22 (0.21–0.22) | 0.10 (0.10–0.10) | 0.18 (0.18–0.19) |
| Significant decrease in water availability due to HI | 0.29 (0.21–0.34) | 0.35 (0.34–0.35) | 0.36 (0.35–0.36) | 0.21 (0.21–0.21) | 0.29 (0.29–0.29) | 0.21 (0.21–0.21) |

HI, human intervention.
Table 2 shows the population-weighted relative location-value of: those being exposed to water scarcity under the HI run; those who moved into/out of water scarcity due to HI; those who already live in water scarcity and experienced a significant aggravation/alleviation of its water scarcity conditions due to HI; those who experience significant increases/decreases in water availability due to HI; and those who experience significant increases/decreases in average duration and occurrence of water scarcity due to HI. The results presented in the table show the ensemble medians per global hydrological model or over the full ensemble, with the interquartile ranges (q25,q75) between brackets.

and seasons. Identifying the main driver (LULCC including the increase of local runoff due to irrigation and reservoirs versus upstream water consumption) of HI impacts as well as its origin (being triggered locally or in upstream areas) reveals that incoming discharge is the dominant origin of HI impacts for

61.8% (46.1–72.0%) of the global population (Fig. 3 and Table 3). In regions being impacted negatively by HI the dominance of incoming discharge is even higher (on average 87.2% of the population), which highlights the dependency of these areas on human actions and decisions being taken upstream. In contrast,

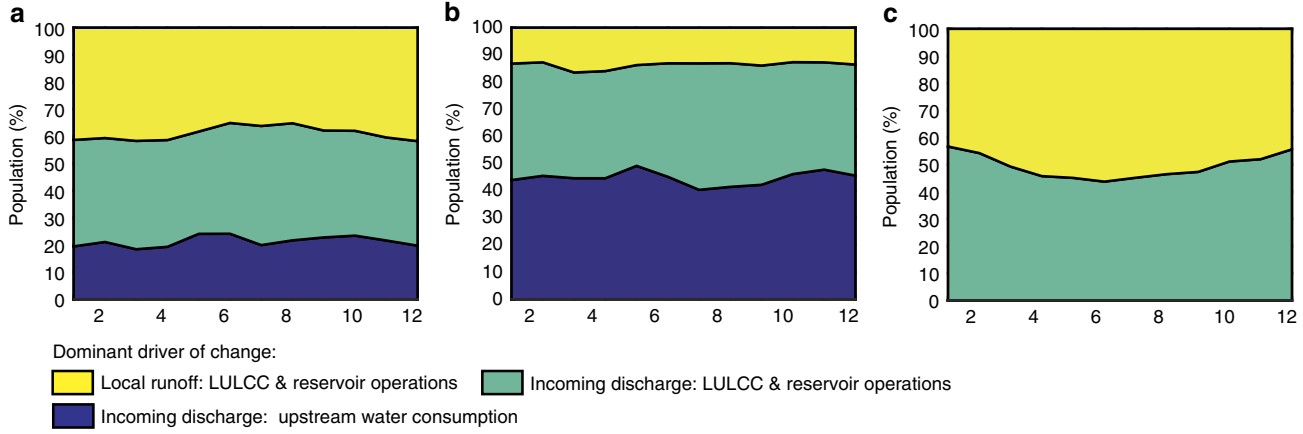

**Figure 3 | Dominant driver of changes in water resources due to HI.** Figure 3 visualizes the share of the global population (2010 values) with a dominant driver and origin of change: Local runoff: LULCC & Reservoirs; Incoming discharge: LULCC & Reservoirs; and Incoming discharge: Upstream water consumption. Figure 3 shows the dominant drivers of change when taking into account all significant changes in water availability due to HI (**a**); significant decreases in water availability due to HI (**b**); and significant increases in water availability due to HI (**c**).

**Table 3 | Dominant driver of significant changes in water availability due to HI and its origin.**

|  | Ensemble-median (WA dec/WA inc) | H08 (WA dec/WA inc) | LPJmL (WA dec/WA inc) | MATSIRO (WA dec/WA inc) | PCR-GLOBWB (WA dec/WA inc) | WaterGAP (WA dec/WA inc) |
|---|---|---|---|---|---|---|
| *Dominant source* |  |  |  |  |  |  |
| Local runoff | 38.2 (12.8/50.6) | 28.0 (0/48.9) | 29.1 (0.8/46.3) | 38.6 (38.0/42.3) | 42.2 (12.6/82.9) | 53.9 (49.4/64.3) |
| Incoming discharge | 61.8 (87.2/49.4) | 72.0 (100/51.1) | 70.9 (99.2/53.7) | 61.4 (62.0/57.7) | 57.8 (87.4/17.1) | 46.1 (50.6/35.7) |
| *Dominant driver* |  |  |  |  |  |  |
| LULCC and reservoir operations | 78.1 (58/100) | 74.5 (40.2/100) | 81.3 (49.5/100) | 67.2 (59.0/100) | 82.6 (68.7/100) | 77.6 (68.1/100) |
| Upstream water consumption | 21.9 (42/0) | 25.5 (59.8/0) | 18.7 (40.4/0) | 32.8 (41.0/0) | 17.4 (31.3/0) | 22.4 (31.9/0) |

HI, human intervention; LULCC, land use and land cover change.
Table 3 shows for the full ensemble and per global hydrological model (GHM) the share of the global population that has local runoff or incoming discharges as dominant origin of changes in water availability due to HI. It also summarizes which share of the population (median values) has LULCC and reservoir operations or upstream water consumption as dominant driver of changes in water availability due to HI. In the results, a distinction is being made between all significant changes in water availability, significant decreases in water availability, and significant increases in water availability and their associated dominant driver of change as well as its origin. The results presented in the table show the ensemble medians per GHM or over the full ensemble, with the interquartile ranges (q25,q75) between brackets.

in regions being positively affected by the implementation of HI (increasing water availability), local runoff acts as the dominant trigger of change (50.6% of the population), highlighting the relative self-sufficiency—and its positive consequences—of these regions in managing their water resources. Reservoir operations and LULCC are the dominant drivers of all HI-driven changes in water availability in regions inhabited by 78.1% (77.6–82.6%) of the global population. At the same time, we find that upstream water consumption is a dominant driver of change in regions inhabited by 21.9% of the global population only. Focusing on the negative impacts of HI only, we find, nevertheless, that upstream water consumption becomes significantly more important as a driver of change, being dominant in regions inhabited by almost half of the global population exposed (42.0%).

The earlier observed global dominance in drivers of change is, however, not spatially uniform (Fig. 4 and Supplementary Figs 7–9). Whereas upstream water consumption is the dominant driver of changes in water availability in May and December for the Huang He and Ganges–Brahmaputra basin, respectively, LULCC and reservoir operations dominate the change in water availability in the Paraná basin year-round (Supplementary Fig. 7). Incoming discharge is relatively more often the dominant origin of change in regions that experience decreases in water availability due to HI (Supplementary Fig. 8), especially when compared with those areas where HI impacts water availability positively (Supplementary Fig. 9). Increases in water availability

are controlled by reservoir operations and LULCC in all river basins, although its dominant origin differs from region to region (Supplementary Fig. 9). If we only examine the influence of reservoir operations and LULCC on the critical dimensions of water scarcity, we find a net decrease in the global population exposed to water scarcity and decreases in the average duration and occurrence of water scarcity for a relatively higher share of the global population, compared to the impacts of all HI together (Supplementary Figs 10 and 11). Nevertheless, still a substantial portion of the population experienced a significant aggravation of its water scarcity conditions, or a movement into water scarcity (Supplementary Figs 12–14), also when we only examine the impact of these two forms of HI.

Compared with those regions with a significant trend in climate change impacts on water resources over 1971–2010, affecting on average 12.1% of the global population, we find, finally, that HI impacts contributed or dominated the change in water resources over time in a significant part of the globe, inhabited by 8.2% and 1.5% of the global population, respectively (Table 4). In addition, HI impacts significantly changed the availability of water resources over time in regions not exposed to a significant trend in climate change impacts, affecting an extra 6.3% of the global population. Also here, significant differences exist in the global distribution of dominant trends and impacts between August and December (Fig. 5). For example, when looking at the relative influence of HI impacts on the availability

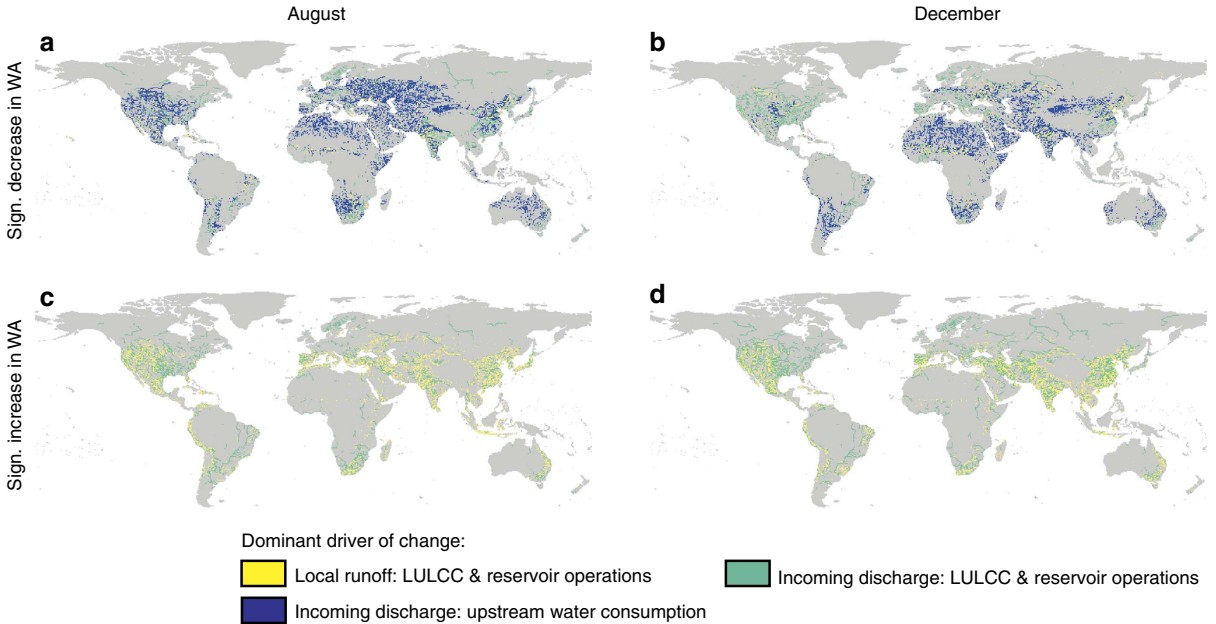

**Figure 4 | Regional and seasonal variation in the dominant driver of change in water availability.** Figure 4 shows for the months August (**a**,**c**) and December (**b**,**d**) the dominant driver of change due to HI, together with its origin: Local runoff: LULCC & Reservoirs; Incoming discharge: LULCC & Reservoirs; and Incoming discharge. Although (**a**,**b**) account only for the significant decreases in water availability (WA), (**b**,**d**) include only the significant increases. The dominant driver of change and its origin are shown only if found to be consistent across a majority of model combinations (>7).

**Table 4 | Relative influence of HI on changes in water availability over 1971–2010 compared with the trend in climate change impacts.**

| | Ensemble-median (q25–q75) | H08 (q25–q75) | LPJmL (q25–q75) | MATSIRO (q25–q75) | PCR-GLOBWB (q25–q75) | WaterGAP (q25–q75) |
|---|---|---|---|---|---|---|
| Sign. climate change impact trend on water availability | 12.1 (7.6–15.6) | 7.5 (6.1–11.0) | 7.0 (6.1–8.6) | 16.1 (13.1–18.0) | 15.0 (13.8–15.6 ) | 10.2 (8.5–15.5) |
| HI contributing to the climate change impact trend | 8.2 (6.2–12.1) | 6.1 (5.2–9.0) | 5.7 (5.1–7.0) | 10.7 (8.8–12.1) | 12.3 (11.5–13.6) | 7.8 (6.7–12.4) |
| HI dominating the climate change impact trend | 1.5 (0.9–3.2) | 1.1 (0.7–1.5) | 0.9 (0.6–0.9) | 5.4 (4.2–5.9) | 1.4 (1.0–1.9) | 2.4 (1.7–3.2) |
| Sign. climate change impact trend only | 0.3 (0.0–0.7) | 0.3 (0.3–0.5) | 0.4 (0.4–0.7) | 0.0 (0.0–0.1) | 0.8 (0.7–0.8) | 0.0 (0.0–0.0) |
| Sign. HI impact trend only | 6.3 (5.4–8.3) | 6.3 (5.8–6.8) | 5.8 (5.5–6.0) | 9.5 (8.5–9.9) | 3.5 (3.4–3.5) | 8.4 (7.3–8.8) |

HI, human intervention.
Table 4 shows for the full ensemble and per global hydrological model (GHM) the share (% of 2010 population values) of the global population exposed to a significant trend in climate change impacts on water availability over the period 1971–2010. Moreover, it shows the share of global population living in areas where HI contribute or dominate this trend. Finally, it shows the share of global population living in areas with only a significant climate change impact trend or only a significant HI impact trend. The results presented in the table show the ensemble medians per GHM or over the full ensemble, with the interquartile ranges (q25, q75) between brackets.

of freshwater resources in India, the Middle East region, Australia or Latin America between August and December.

**Sensitivity to the choice of model and forcing data**. The GHMs used in this study show a relative constant modelling spread in outcomes for each of the analysis performed (Tables 1–4). Whereas MATSIRO presents the high-end outcomes in all cases, the result of the other GHMs are more closely related, for example, with respect to the long-term mean exposure to water scarcity events or the increase in average duration and occurrence of water scarcity due to HI. Compared with the other models, MATSIRO also shows to have the highest sensitivity to the use of different forcing data sets, as shown by the relatively large interquartile ranges presented for each of its outcomes.

When looking at the impact of HI we find that almost all results per GHM show to be significantly different from zero,

irrespectively of the GHM used (Table 1). The impacts of HI are globally significant, both with respect to the changes in water availability, as well as regarding the changes in exposure to water scarcity (movement in/out, aggravation/alleviation), and the changes in average duration and occurrence of water scarcity events. Only the estimated net changes in exposure to water scarcity as a result of HI show ambiguous outcomes, with the interquartile ranges varying from negative to positive values. This can be explained by the variation in estimated net effects between the different GHMs studied and forcing data sets used. For a majority of GHMs (PCR-GLOBWB, WaterGAP and MATSIRO), we find, on average, a relative higher share of the global population that experienced significant decreases in water availability, an alleviation of water scarcity conditions, a movement into water scarcity or an increase in the persistence of water scarcity events due to HI. H08 and LPJmL, on the other hand, show a relative higher share of the global population for

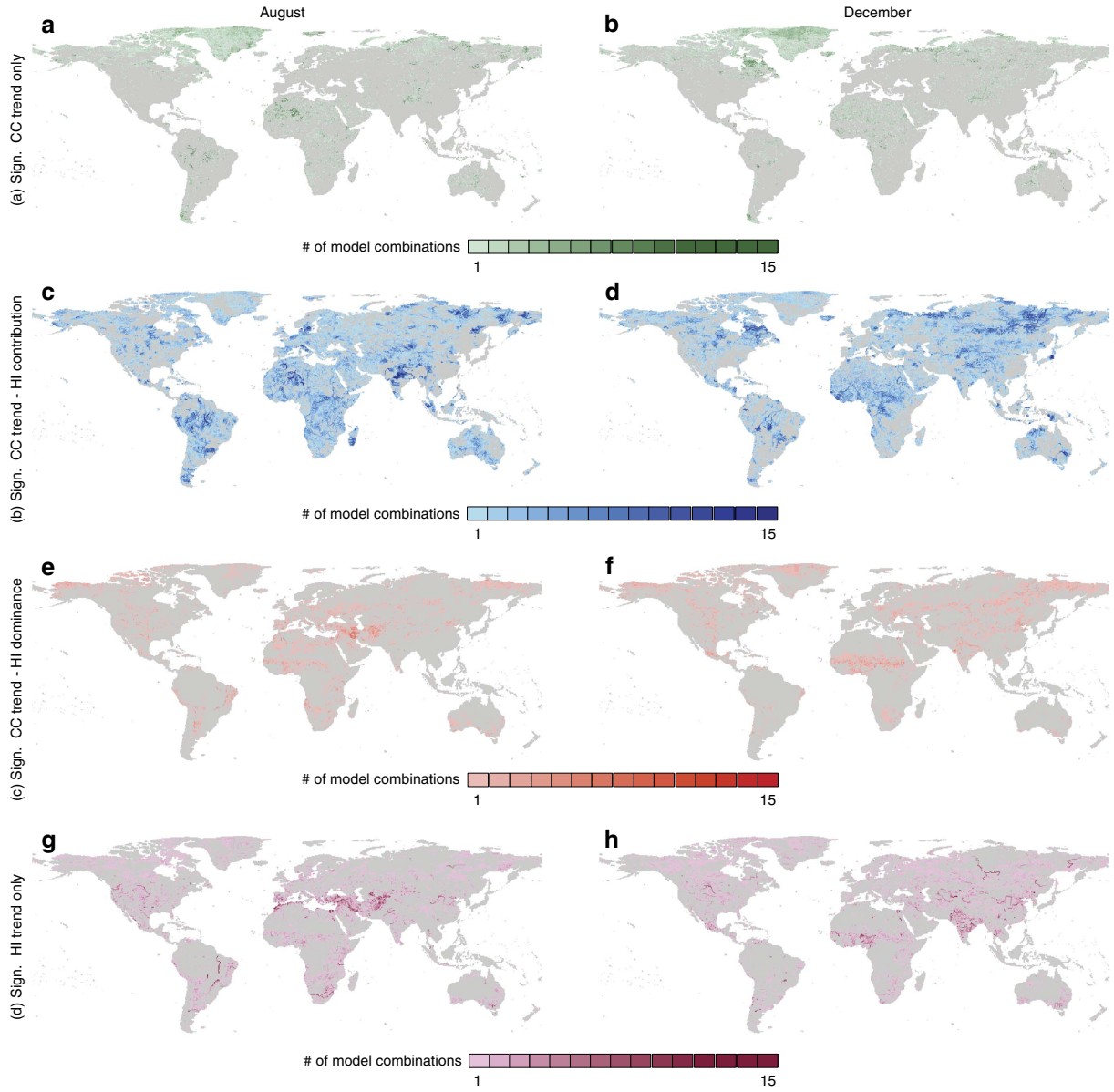

**Figure 5 | Relative influence of HI compared with the trend in climate change impacts.** Figure 5 shows the spatial and seasonal variation in the relative influence of HI as driver of changes in water availability over the period 1971–2010, compared with the climate change impact trend. Here we show per area the number of model combinations with: a significant climate change impact trend on water availability only (**a,b**); HI contributing to the climate change impact trend on water availability (**c,d**); HI dominating over the climate change impact trend on water availability (**e,f**); and a significant HI trend only (**g,h**). Results are shown separately for the months August (**a,c,e,g**) and December (**b,d,f,h**).

which HI caused an increase in water availability, an alleviation of water scarcity conditions, a movement out of water scarcity or a decrease in the persistence of water scarcity events.

All GHMs depict changes in incoming discharges to be the largest source of decreases in water availability due to HI (Table 3), thereby providing robustness to the ensemble-median result. Local runoff as the source of change becomes significantly more dominant in all GHMs when looking at the increases in water availability due to HI only, with PCR-GLOBWB and WaterGAP, indicating here the changes in local runoff to be dominant in a slight majority of the globe. Reservoir operations and LULCC were perceived by all GHMs to be the dominant driver of changes in water availability due to HI. Although reservoir operations and LULCC drive the increases in water

availability due to HI in all GHMs, they dominate the decreases in water availability for a majority of the globe in LPJmL, MATSIRO, PCR-GLOBWB and WaterGAP. Only in H08, upstream water consumption becomes a dominant influence in a majority of the globe when looking at the decreases in water availability due to HI. As reflected by the ensemble-median results, upstream water consumption becomes, nevertheless, in each of the GHMs a significantly more important driver of change when looking at the decreases in water availability only compared to all changes in water availability.

Each of the GHMs indicate, finally, that HI contribute significantly to the trend in climate change impacts over the past 40 years, affecting 5.7% (LPJmL) to 12.3% (PCR-GLOBWB) of the global population (Table 4). At the same time, we find that

HI dominate over the climate change impacts trend in regions, inhabiting 0.9% (LPJmL) to 5.4% (MATSIRO) of the global population. These cross-model variations can be explained partly by the differences in areas with a significant climate change impact trend. Whereas only 7–7.5% of the global population experienced a significant change in climate change impacts over time according to H08 and LPJmL, results are substantially higher for PCR-GLOBWB (15.0%) and MATSIRO (16.1%).

## Discussion

This research is the first to provide insights on the impacts of HI (individually and altogether) on water scarcity over time, by evaluating how HI have influenced the critical dimensions of water scarcity, whether and to what extent they have led to a reshuffling of water scarcity hotspots and who are winners or losers. Using time-varying information on dam and reservoir operations, LULCC and water demands, our study identified for the first time the dominance of different drivers and sources of change, and compared them with the trend in climate change impacts. By introducing a spatially and temporally explicit measure of minimum environmental flows, combined with the inclusion of seasonality, our water scarcity estimates are more realistic and significantly lower than the monthly results presented in previous studies[17,58]. Although Wada et al.[17] and Mekonnen et al.[58] applied spatially and temporally uniform environmental flow requirements and assumed that all people who experience water scarcity experience it at the same point in time, our results show a clear seasonal pattern for the majority of the river basins studied. It must be acknowledged, however, that this seasonal approach limits our analysis to the fluxes (runoff and discharge) of the hydrological cycle, whereas a long-term equilibrium was assumed for the storage components (lakes, aquifers and sub-surface reservoirs). In this study, we assume that all water is taken out from the river and that water users do not have the opportunity to take water from the lakes and groundwater. As a result we underestimate water availability in those regions that heavily rely on these storage components and we therefore expect the water scarcity numbers to be lower in reality. The multi-model framework of this study allowed for more robust estimates compared to previous studies executed in a single-model setting[7–11,13–17].

Our results demonstrate that HI implemented throughout the period 1971–2010 have led to substantial changes in the critical dimensions of water scarcity and significantly reshuffled global water scarcity hotpots. Although the net impact of HI on the human exposure to water scarcity appears limited at the global scale, we show that large spatial differences underlie this result. In most regions, substantial parts of the population face aggravated water scarcity conditions due to HI or move into water scarcity, although others encounter alleviations or move out of water scarcity. We also find that HI increase the average duration of water scarcity events affecting a significant share of the global population.

Analysis of the dominant sources and drivers of changes in the availability of water resources highlights that those regions negatively affected by HI are located on average further downstream in river basins than areas being impacted positively, whereas they rely predominantly on actions taken upstream. This leads globally to water scarcity travelling downstream due to HI. The found spatial and seasonal variation in the dominant driver and sign (positive/negative) of HI impacts illustrate, however, that HI attribution to changes in water availability and water scarcity is complex, and that the found relations should not be considered to be homogeneous through time. This emphasizes a thorough consideration of such interactions throughout the process of design and implementation of water scarcity adaptation. The

modelling spread that underlie our results is, moreover, a call for the different global models to further investigate the incorporation of HI in their modelling framework, thereby accounting for possibly interaction effects and feedback linkages.

Foreseen changes in climate change and socioeconomic developments[18–28] are expected to amplify the observed trends and to reinforce the observed pattern of winners and losers. HI may alleviate/aggravate, on the one hand, the vulnerability to climate change and climate variability, especially in those regions that experience significant increases/decreases in water availability due to HI. Climate change, climate variability and changes in local water demands, on the other hand, may significantly strengthen/weaken the HI impacts in future time periods by putting more/less pressure on the water resources. Our results show a significant trend in hydro-climatic impacts on water availability over the period 1971–2010 for a substantial part of the globe, with on average 12.1% of the global population being exposed. HI impacts significantly contributed to or even dominated these historical trends in regions inhabiting 9.7% of the global population. In addition, 6.3% of the global population experienced significant changes in water resources over time due to HI only. Being sensitive to changes in either climate change, HI and/or local water demands can have substantial implications for the choices being made in the design of adaptation strategies to cope with current and future water scarcity[18–28]. For example, by focusing on decreasing water demands via technical (for example, increasing efficiency), regulatory (for example, quota) or market-based measures (for example, water pricing), by targeting the increase of water supply (for example, desalination, reservoirs, water transfers and renewable groundwater) or by making sure water resources are being allocated in an equitable, sustainable way (for example, treaties). Such water management decisions should not be taken in isolation, as our study shows that the consequences of upstream measures and activities can be far-reaching in downstream areas. Coping with local problems related to the ongoing increases of human pressures on water resources therefore requires a higher level—regional or global—understanding of both the underlying mechanisms and drivers, their origin and their potential effects[16].

Apart from that, economic impacts, costs, but also societal perception and political willingness in decision-making processes determine greatly the local adaptation space to cope with water scarcity and often prioritize focus areas and actions being undertaken. Its complexity (incorporation of feedback loops between nature and society, and vice versa), its specific model requirements (coupling GHMs with economic welfare models, partial-equilibrium models or agent-based models) and its specific information needs (behavioural economics) require a thorough development of our global modelling principles and involve extensive research. This prohibits—at this moment—such a study at the global scale. Given its importance and necessity, we do encourage research to take this direction to facilitate studies to any of these human-nature interactions and its societal impacts in a fully integrated manner.

## Methods

**Testing the impact of HI on water scarcity.** In this study we evaluated how time-varying HI affected water availability and water scarcity conditions in the late twentieth and early twenty-first century, how they influenced the critical dimensions of water scarcity and how they reshuffled water scarcity hotspots. To this end, we performed a scenario analysis to compare gridded (0.5° × 0.5°) monthly estimates of water availability and water scarcity conditions from the NHI run with those from a (time-varying) HI (HI) run, including the impacts of LULCC, reservoir and dam operations and upstream water consumption, altogether and separately.

**Hydrological data under the ISI-MIP Phase 2 framework.** An ensemble of 15 model combinations (5 GHMs forced by 3 observation-based historical climate

data sets, Supplementary Table 1) was used to evaluate for each ensemble member individually the impact of HI on water availability and water scarcity. Subsequently, we calculated ensemble median values and presented these results together with the modelling spread. The GHMs used in this analysis are: H08[41,42], LPJmL[43,44], MATSIRO[45], PCR-GLOBWB[46,47] and WaterGAP[13]. Each of these GHMs was forced with daily/three hourly (MATSIRO) inputs from three observations-based historical climate data-sets: PGFv2 (ref. 48), GSWP3 (http://hydro.iis.u-tokyo.ac.jp/GSWP3) and WFD/WFDEI[49] under the framework of ISIMIP2a (phase 2 of the Inter-Sectoral Impact Model Intercomparison Project, www.isimip.org). For a comprehensive overview of the different historical climate data sets, we refer to the individual references and to Müller Schmied et al.[13].

Water availability refers here to the monthly availability of water in rivers, which consists of the locally generated runoff in each $0.5° \times 0.5°$ grid cell and the incoming discharge from upstream cells being diminished with the upstream water consumption[17]. In doing so, we only account for the water fluxes (streamflow, (sub-)surface runoff and baseflow) in our analysis and assume a long-term equilibrium in the storages of the hydrological system (such as lakes, aquifers and sub-surface reservoirs), for which the absolute values are unknown. Using the total monthly water availability per cell under pristine conditions, we subsequently estimated the minimum environmental flow requirements[54–57] based on the variable monthly flow methodology[57], see Supplementary Methods. The consumptive water use that are used by the GHMs to adjust the streamflow under the HI run and the water withdrawals to assess water scarcity conditions under NHI and HI run encompass water demands of the agricultural sector (irrigation and livestock), the industry sector (thermal energy and manufacturing) and water demands for domestic use, and are modelled using various socio-economic parameters (GDP, population density, livestock density, land use and land cover)[2]. The net amount of incoming discharge (being accounted for the consumptive water needs) varies, thereby across the GHMs, not only due to differences in generation of discharge or the height of the modelled water demands but also due to variations in the allocation of the consumptive water needs over surface and/or groundwater. The combination of these aspects are ground for the observed variation in water availability across the GHMs, the differences in the exposure to water scarcity and, subsequently, the impacts of HI on freshwater resources and the critical dimensions of water scarcity. The methods used to model water availability, to estimate water demands, and to allocate consumptive water demands over surface and groundwater resources are summarized in the Supplementary Methods and for a more extensive discussion we refer to Wada et al.[2] or the individual model refs 13,41–47.

**Assessing water scarcity.** Water scarcity conditions were assessed in this study by means of the WSI[1]. The WSI estimates the ratio between water withdrawals and water availability for humans, and is extensively used in water scarcity assessments at global and regional scales[17–20,22,24,25,58,60–64]. In this study we follow Mekonnen et al.[58] and explicitly incorporate minimum environmental flow requirements when estimating water scarcity conditions. Using the WSI, a region is considered to experience water scarcity if $WSI_{i,m} > 1$, that is, if > 100% of the total available water resources is being allocated for environmental and anthropogenic needs[58]:

$$WSI_{i,m} = \frac{WW_{i,m}}{Q_{i,m} - EF_{i,m}}, \qquad (1)$$

where $WSI_{i,m}$ is the WSI for cell $i$ and month $m$, $WW_{i,m}$ is the total water withdrawal in cell $i$ and month $m$, $Q_{i,m}$ the total river water availability in cell $i$ and month $m$, and $EF_{i,m}$ the environmental flow requirement[57]. In our analysis we underestimate the actual amount of water that may be available from groundwater, reservoirs and lakes at the monthly scale, and therefore overestimate water scarcity where these water sources are used for water supply.

**Incorporating HI in the modelling framework.** Three types of HI were included in this study: LULCC, dam and reservoir operations and upstream water consumption. The HYDE 3—MIRCA data set[50–52], assembled following Fader et al.[53], was used by each of the GHMs, apart from WaterGAP, for simulating the effects of changes in irrigation and/or cropland patterns on the generation of local runoff. Although land-use conversions to agricultural land use tend to increase the volume of water that runs off and increases the speed of runoff processes, irrigation water use generally decreases runoff due to increased evapotranspiration rates[15]. The GranD database[65] was included in all GHMs to represent the historical development in dams and reservoirs. Dam and reservoir operations shift downstream streamflow patterns and decreased seasonal flow amplitudes, especially if built for energy generating purposes. Moreover, dams and reservoirs affect the absolute volumes of water via enhanced evaporation losses and direct abstractions. Each of the GHMs distinguishes dams and reservoirs built for irrigation and/or non-irrigation purposes, whereas PCR-GLOBWB additionally identifies flood control and navigation purposes. The reservoir operation schemes that are applied in each of the GHMs are based on Hanasaki et al.[66] (H08, MATSIRO and WaterGAP), Biemans et al.[8] (LPJmL) and Haddeland et al.[10] in combination with Adam et al.[67] (PCR-GLOBWB). Whereas the operation schemes of Hanasaki et al.[66], Biemans et al.[8] and Haddeland et al.[10] are retrospective, which ensures optimal performance given its purpose, inflow and demand, PCR-

GLOBWB implemented a prospective scheme that has to deal with uncertain forecasts of supply and demand[46]. All models, apart from H08, accounted for increased evapotranspiration over reservoirs. For a more detailed discussion on the parameterization of the reservoirs within each of the GHMs, we refer to the specific model references[13,41–47]. Demand growth and its impact on water availability and water scarcity was covered by the inclusion of the net upstream water abstractions (that is, withdrawal—return flow) from the streamflow, as calculated by each of the GHMs. Historical demand growth was also evaluated by inclusion of the historical trends in local water withdrawals, influencing the water scarcity conditions. For a detailed discussion on the inclusion of these HI in the modelling framework of each of the GHMs, we refer to the Supplementary Methods and the individual model references[13,41–47].

**Quantifying the impacts of HI.** To assess the impacts of HI on the monthly water availability and water scarcity conditions we performed a scenario analysis with two simulation runs. In the first run (NHI), we evaluated water availability, water scarcity conditions and exposure to water scarcity events without HI on the streamflow. In the second run (HI), we evaluated water availability, water scarcity conditions and exposure to water scarcity events including the impact of HI. In an additional analysis, we separated the impacts of demand growth via upstream water consumption from the impacts of the other HI (dam and reservoir operations and LULCC).

Impacts of HI on the availability of water resources and the critical dimensions of water scarcity were evaluated for each ensemble member individually. Thereafter, ensemble-median values were calculated and presented together with the interquartile range. All model combinations are weighted equally: a weighting based on the performance of the individual forcing datasets or GHMs was not executed. Impacts of HI are expressed by % of the population exposed (using 2010 values) at the global and regional scale looking at significant changes in the critical dimensions of water scarcity (average duration, occurrence and severity of water scarcity), in the availability of water resources and by showing who is moving in/out of water scarcity. The relative location of impacts in river basins was estimated by comparing the upstream area of a specific impact location (that is, experiencing a significant increase or decrease in water availability due to HI) with the total river basin area. For example, a value of 0.5 refers here to a location for which the upstream area is half of the total river basin area, whereas a value of 1 refers to the outlet of a basin into the ocean (or an internal sink) (that is, the size of the upstream area is equal to the size of the total river basin). To show the global- or basin-mean relative location of impacts, we aggregated these location values by using population-weighted means. In all analyses, only changes > |5%| were considered to be significant and taken into account. Finally, we evaluated for each month the trend in HI impact on water availability over the period 1971–2010 using linear regression analysis. We compared the trends in HI impacts with the trend in impacts of climate change on the availability of water resources. In doing so, we indicate where and when HI buffered, strengthened or even dominated the hydro-climatic impacts. When interpreting the results of this analysis, one should take into account, however, that the climate data used in this study, and particularly its length, is not fully suitable for trend analysis.

**Testing the sensitivity of results.** Using an ensemble of 15 model-combinations (5 GHMs and 3 forcing datasets) to assess the impacts of HI on water scarcity conditions, the exposure to and average duration of water scarcity events, and the underlying changes in water availability, allows not only to make more robust estimates compared with a single-model study, it also enables the evaluation of modelling uncertainties. The results presented here in graphs of the main body of text concern the ensemble-median (q50) values together with their interquartile ranges (q25–q75). Gridded results were presented if the found signal was consistent and significant for a majority (>7) of the model combinations, otherwise the number of model combinations with a significant signal was visualized.

**Data availability.** The hydrological data sets used in this study are generated under the framework of phase 2 of the ISI-MIP project and will become soon publicly available via www.isimip.org. The historical climate data sets that have been used to force the GHMs (PGFv2, GSWP3 and WFD/WFDEI), the HYDE3—MIRCA data-set describing the historical changes in land use and land cover, and the GranD database that describes the historical development in dams and reservoirs can be downloaded from www.isimip.org.

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

## Acknowledgements

This work has been conducted under the framework of ISIMIP2a (www.isimip.org) and we want to thank the coordination team responsible for bringing together the different global hydrological modelling groups and for coordinating the research agenda, which resulted in this manuscript. T.I.E.V. was funded by the EU 7th Framework Programme through the projects ENHANCE (grant agreement number 308438) and Earth2Observe (grant agreement number 603608). J.C.J.H.A. received funding from the Netherlands Organisation for Scientific Research (NWO) VICI (grant number 453-14-006). Y.M. was supported by the Environment Research and Technology Development Fund (S-10) of the Ministry of the Environment, Japan. P.J.W. received funding from the Netherlands Organisation for Scientific Research (NWO) VIDI grant (number 016.161.324). H.K. and T.O. were supported by Japan Society for the Promotion of Science KAKENHI (16H06291). J.L. received funding from the National Natural Science Foundation of China (41625001).

## Author contributions

T.I.E.V., J.C.J.H.A., Y.W. and P.J.W. designed the research. T.I.E.V., P.D., S.N.G., J.L., Y.M., H.K., T.O., S.O., Y.P., Y.S. and Y.W. performed research. T.I.E.V., J.C.J.H.A., Y.W. and P.J.W. analysed data. T.I.E.V. wrote the paper. S.N.G. coordinated impact modelling experiments. All authors provided comments and guidance on the manuscript.

## Additional information

**Competing interests:** The authors declare no competing financial interests.

**Publisher's note**: 

