## [Peer Review File · Nature Communications]

Reviewers' Comments:

Reviewer #1 (Remarks to the Author)

Global water scarcity is one of the most pressing environmental and natural resource challenges we face. The submitted article considers the effects of human interventions (defined by the authors to include consumptive water use plus various other interventions) on patterns of global water scarcity. Its subject matter is therefore entirely suitable for Nature Communications. The modeling is performed on a monthly timescale, a great attribute given that most rivers exhibit tremendous seasonality, so the seasonal timing of intervention effects is crucial to assessing scarcity. The paper provides a quantitative demonstration that human interventions in natural watersheds shift water scarcity problems spatially, benefiting some and taking away from others, and creating the scarcity "hot spots" the authors note in their title.

My recommendation is for publication pending major revisions. The manuscript has very good potential for publication in Nature Communications, but requires improvement before that should happen. I hope the following comments will prove helpful to the authors as they continue to work on their article.

Though in some respects the submission appears novel (more on this shortly), the basic idea that (for example) building a dam and withdrawing water can create winners in one part of the basin and losers in another, especially for transboundary rivers, is not. It would be interesting to see how this compares to the body of work assembled by social science-oriented workers like, say, Wolf or Bohmelt. One point of interest is why certain large basins show a more inequitable distribution of intervention-induced water shortages than others, a pattern that is quite clear in the authors' outcomes. Or to put it another way, the paper identifies a number of "hot spots" but doesn't delve into why these spots are hot and others are not. Also, a number of modeling studies have considered the impacts of demand growth and climate change on global patterns of water scarcity. I think the article could do a better job of specifying how its methods and outcomes are different from that prior work. The explanation doesn't need to be lengthy, but it does need to be precise.

Further, as the paper is currently written, the language around the relationships between various drivers of change seems quite vague. For example, it's unclear whether and if so how demand growth is separated out from other human interventions. If demand growth is included in the human interventions here, then how much of the total change in scarcity is due to this obviously huge driver vs. construction of dams and land use/land cover change? (Expressing the results as a percentage of the population experiencing water stress, which is done at a few points in the paper, helps normalize the result for population growth but not for other demand drivers.) Does the relative importance of various contributions vary spatially? And how does it tie in to the "hot spots" the authors identify?

To elaborate further on this point: a clearer explanation of which human interventions were included, and how they were incorporated into the model, is needed. The study takes the

commendable step of using multiple hydrologic models and climate forcing datasets, but it's not clear how uncertainty in human interventions was incorporated into the study, if at all. It's also not clear how the hydrologic impacts of LULC (say, higher peak runoff and lower minimum runoff associated with urbanization) were incorporated, or those of reservoir operations (and associated rule curves for releases, etc, which can shift downstream seasonal flow patterns and thus seasonal water scarcity patterns, even without affecting total availability, particularly for dams focused on flood control or hydropower). Though the paper directly states that it includes effects from LULC and reservoir operations, I only see detailed discussion of consumptive use (demand) effects. Demand changes may be associated with LULC change and reservoir operations, but these are fundamentally different things. By the same token, it's great that the authors used monthly streamflow simulations to capture seasonal dynamics, but how did the models incorporate reservoir operation to shift water supply from low-demand months to high-demand months, as is routine water management practice?

Along similar lines, discussion of climate variability and change is virtually absent from the paper. I acknowledge that this is not the main topic of the paper, and that by comparing intervention and non-intervention runs, the effects of climate change are at last partly accounted for. But this still seems like a gap in the analysis. For example, the authors should be able to use a comparison between the intervention and non-intervention runs to estimate how the magnitude of impacts from interventions compare to the impacts of climate change over the 40 year simulation period. It would also be useful to see a few words about how the impacts of interventions may alleviate or exacerbate water scarcity vulnerability to climate change (see recent Hydrol Sci J paper by Fleming). My guess is that the pattern of winners and losers created by interventions, as outlined in the submitted manuscript, might be reinforced by the seasonal flow timing changes expected for most parts of the world under climate change.

On a more technical note, the instream flow requirement definition used by the authors doesn't appear to be widely employed (I've never seen it before, and it looks like only one reference is cited), yet it forms an important part of their water scarcity index. IFN is a large multi-disciplinary field encompassing multiple approaches. In the supplementary materials, I'd like to see a little more explanation provided for the means used to estimate environmental flow needs. This is a question that seems sure to arise for readers who are specialists in areas like hydroecology, habitat assessment, aquatic species diversity, and so forth.

Overall, then, while this paper holds substantial promise, there are a lot of unanswered questions about both methodology (which goes to experimental reproducibility) and implications (which goes to relevance) that need addressing before the article is ready for publication.

There are also a few, more minor, points that the authors might address to improve the readability of their article. For example, explanations of several of the figures would benefit from some further work. Figures 2, 3, and 4 all show similar, but not the same, quantities. More direct statements of the punch lines from each of these figures (possibly just within the figure captions themselves) would be helpful to the reader. It would also be nice to add a version of Figure 6 giving the same analysis product, but for the non-intervention run. This would enable the reader to more easily tease out intervention effects from climate effects over the multi-decadal model run. The content of

Figure 7 could use some clearer explanation; I think this ties back to around line 159 of the manuscript, where a distinction is drawn between sources of change lying upstream of the sub-basin, within the sub-basin, and locally, but without explicitly saying how sub-basins are defined and what exactly is the difference between “within the sub-basin” and “locally.” The reader can make an educated guess, of course, but the ambiguity shouldn’t be there.

The terminology could also be improved. For example, I strongly suggest using a word other than “persistence” to describe the duration of water scarcity events as defined in this paper. Persistence has a very specific meaning in hydrology and climatology, referring to statistical descriptors of system memory (serial correlation, AR processes, Box-Jenkins models, Hurst effect, etc). I suspect there’s a tie between this meaning and what the authors mean by it, but they’re not the same. Perhaps “event duration” would be better.

Similarly, there is a small but unnecessary inconsistency between the definitions for “blue water” provided in the main article versus that in the first paragraph of the supplementary material. Groundwater is included as blue water in the former, but as green water in the latter. Further, this blue vs. green water distinction, though somewhat popular in global water resource studies (particularly in policy papers by social scientists, where perhaps it has some legitimate descriptive use), is fundamentally dubious at an Earth science level and rarely if ever appears in high-quality local or regional scale hydrologic modeling and water resource planning work. Soil moisture and groundwater (so-called green water) are major physical controls of river flows and lake levels (so-called blue water), and are therefore directly represented (in various fashions) in watershed models simulating surface water resources. Separating the two makes little scientific sense in a modeling study and does not appear to serve a useful function in this article.

Reviewer #2 (Remarks to the Author)

The paper is well written and very interesting for the reader/community who are particularly working on water scarcity topic. It analysis the impact of human interventions on water scarcity over the period of 1971-2010. The research done for the paper has several novel aspects: I particularly appreciated inclusion of environmental flow requirements and non-human intervention comparisons. The method followed were well described and based on solid scientific background. There are some parts which needs improvement and/or clarification, which are indicated below. If the authors can clarify/change them, the paper will have novel aspects for publication.

- Do the authors consider water withdrawal or water consumption for the upstream human intervention? In the abstract, it says water withdrawal but in later stages they mentioned water consumption. It is confusing. This is fundamentally important on water scarcity analysis as upstream water withdrawal may not necessarily affect water availability at the downstream (e.g. return flow and discharge).

- It would be better to explain at the discussion part what does really water scarcity mean. As it is calculated, it actually means reduction of water that is required for environment, does not mean that population experiences any shortages or scarcity. Otherwise exposure to population to water scarcity might be understood differently by readers. Also I questioned if it makes sense to do

analysis of "population exposed to water scarcity" since $WS > 1$ can mean either population and/or environment is exposed to WS: an explanation is required to justify this analysis.

- I found the main conclusion written by the authors - "adaptation measures should be embedded in integrated river basin management plans, addressing upstream effects on downstream water scarcity" - weak. I mean this is well known by water resources management (both scientists and practitioners) and the authors' analysis is not needed to draw this conclusion. It is better if the authors include conclusion that uniquely taken out from their study and revise this policy conclusion statement in this current form.

Reviewer #3 (Remarks to the Author)

The goal of this study is to evaluate the impacts of human interventions on water resources and scarcity. This is an important research question and the authors have made excellent headway in terms of linking the hydrology models to address this research question. However, when understanding the implications of human activities, it is necessary to model relevant human processes explicitly, rather than inputting human actions as simple geographical grids and proxying for human demands with crude correlations. This is the main reason that I do not find this paper suitable for publication at this time. Additionally, climate information is used to drive pretty much all aspects of this study. Climate is an input variable to the HYDE 3.1/MIRCA datasets used by this study, as well as the socio-economic proxies and hydrology models. This provides climate with a far too prominent role in the methodological framework, attributing most human "interventions" to climate drivers. Instead, human activities that are distinct to climate (not all human actions are driven by climate, but by economics, rational/irrational decision-processes, policies, etc.), but impactful to water resources, should be considered and explicitly modeled.

Major comments:

1. Human processes are not explicitly modeled.

The physical variables are modeled very effectively. A state of the art approach was used, which is an ensemble of 15 model combinations (5 global hydrology models forced by 3 climate sets) (line 216). However, human processes are not modeled at any point in this paper. Instead, a set of socio-economic proxies is used to estimate human water demand. In fact, "to estimate irrigation, livestock, and industrial water demand, hydroclimatological parameters are used as well" (SI, line 96). So, this modeling framework solely relates demand to socio-economic proxies (which need to be explained better in the SI since these are foundational to the paper) and hydro-climatological parameters. Concepts of supply and demand are foundational in economics. However, this study of water supply and demand does not take economic principles into account. Supply and demand should be impacted by price, not just climate variables. I understand that this is common practice in the water resources and hydrology literature. But, given that the main goal of this paper is to explore

inter-linkages between human actions and water scarcity, I find it necessary to move the methodology forward and explicitly incorporate these key economic concepts, or at least some explicit social processes to the modeling framework.

2. Irrigation input is fixed in time.

The paper seeks to evaluate human impacts over the period 1971-2010. However, the MIRCA2000 irrigation data set -- which is fixed to the time period around the year 2000 -- was used to force the hydrology models. This is problematic. Additionally, the HYDE 3.1 database was used (line 226) but it is not clear if an annual grid was obtained from HYDE 3.1 or some other temporal average was used for this geographical input. According to the SI, line 101: "All impact models modelled irrigation water consumption and withdrawals endogenously, using the dynamic HYDE 3.1/MIRCA dataset as base-map for the extent of irrigated areas." However, the references provided for these databases on line 226 are references 38, 39, and 40. From reference 39 the MIRCA2000 database is fixed in time. So, it is not clear why these datasets are called "dynamic". This needs to be more clearly explained in future revisions.

2. Groundwater interactions are not clear.

The study claims to evaluate water availability from groundwater sources (line 230). However, none of the results of the paper highlight groundwater resources, which should dampen water scarcity outcomes. All results are given in terms of surface watersheds. The ability to access groundwater resources would enable populations to minimize their vulnerability to local water scarcity in both space and time. It is not clear if this study enables groundwater pumping, and, if so, how and when groundwater pumping interacts with water scarcity outcomes. One of the main conclusions of the study is that upstream water withdrawals lead to significant downstream reductions in water availability (line 132). It is important to understand if these downstream pixels are able to adapt and counteract this by pumping groundwater.

Minor comments:

1. According to the SI, line 106: "Environmental flow requirements were reduced according to the actual irrigation water consumption, while WSI was calculated using the potential irrigation water withdrawals". This seems inconsistent.

2. Exposure of humans to water scarcity. This is done by co-locating human population pixels with quantified values of water scarcity across all economic sectors. It seems more reasonable to quantify populations that live in locations that have high water stress due to only domestic demands outstripping local water availability.

Reviewer #1:

Global water scarcity is one of the most pressing environmental and natural resource challenges we face. The submitted article considers the effects of human interventions (defined by the authors to include consumptive water use plus various other interventions) on patterns of global water scarcity. Its subject matter is therefore entirely suitable for Nature Communications. The modeling is performed on a monthly timescale, a great attribute given that most rivers exhibit tremendous seasonality, so the seasonal timing of intervention effects is crucial to assessing scarcity. The paper provides a quantitative demonstration that human interventions in natural watersheds shift water scarcity problems spatially, benefiting some and taking away from others, and creating the scarcity "hot spots" the authors note in their title.

My recommendation is for publication pending major revisions. The manuscript has very good potential for publication in Nature Communications, but requires improvement before that should happen. I hope the following comments will prove helpful to the authors as they continue to work on their article.

We thank the reviewer for the positive and thorough comments and are pleased that he/she values the scientific relevance of our research. The reviewer provides several very useful comments/suggestions for revisions. We will address these in the revised manuscript, as per our responses to each comment below.

Major comments:

1.1 Though in some respects the submission appears novel (more on this shortly), the basic idea that (for example) building a dam and withdrawing water can create winners in one part of the basin and losers in another, especially for transboundary rivers, is not. It would be interesting to see how this compares to the body of work assembled by social science-oriented workers like, say, Wolf or Bohmelt. One point of interest is why certain large basins show a more inequitable distribution of intervention-induced water shortages than others, a pattern that is quite clear in the authors' outcomes. Or to put it another way, the paper identifies a number of "hot spots" but doesn't delve into why these spots are hot and others are not. Also, a number of modeling studies have considered the impacts of demand growth and climate change on global patterns of water scarcity. I think the article could do a better job of specifying how its methods and outcomes are different from that prior work. The explanation doesn't need to be lengthy, but it does need to be precise.

Many thanks for this suggestion. We highly appreciate the suggestion to relate our results to the work of the more social science oriented workers and did so accordingly (see introduction section of revised manuscript). Moreover we tried to emphasize the novelty of our work, with respect to previous studies, more explicitly. In preparing the revised manuscript, we delved finally more into the underlying mechanisms causing changes in fresh water resources and exposure to water scarcity as a result of human interventions and/or climate change, trying to clarify why certain areas can be identified as hotspots whereas impacts are minor in others (see also our response to 1.2, 1.3, and 1.4).

1.2 Further, as the paper is currently written, the language around the relationships between various drivers of change seems quite vague. For example, it's unclear whether and if so how demand growth is separated out from other human interventions. If demand growth is included in the human interventions here, then how much of the total change in scarcity is due to this obviously huge driver vs. construction of dams and land use/land cover change? (Expressing the results as a percentage of the population experiencing water stress, which is done at a few points in the paper, helps normalize the result for population growth but not for other demand drivers.) Does the relative importance of various contributions vary spatially? And how does it tie in to the "hot spots" the authors identify?

Thank you for this fruitful comment, we believe that this is indeed a valuable extension to our current analysis. Following the Reviewer #1's suggestion to specify the relationship between various drivers of change and to

discuss their impacts on fresh water resources and water scarcity we executed in the preparation of the revised manuscript the following new analysis:

- We separated the impact of demand growth on fresh water resources from those of the other interventions (reservoirs and land use and/or land cover change). The outcomes of this analysis enabled us to individually discuss both the effects of these interventions and the demand growth as well as the interplay between those mechanisms that results in the simulated impacts on fresh water resources;

Using the outcomes of this analysis we tried to clarify the observed pattern of change (regions being hotspots of change or not) in the results and discussion section of the revised manuscript.

1.3 To elaborate further on this point: a clearer explanation of which human interventions were included, and how they were incorporated into the model, is needed. The study takes the commendable step of using multiple hydrologic models and climate forcing datasets, but it's not clear how uncertainty in human interventions was incorporated into the study, if at all. It's also not clear how the hydrologic impacts of LULC (say, higher peak runoff and lower minimum runoff associated with urbanization) were incorporated, or those of reservoir operations (and associated rule curves for releases, etc, which can shift downstream seasonal flow patterns and thus seasonal water scarcity patterns, even without affecting total availability, particularly for dams focused on flood control or hydropower). Though the paper directly states that it includes effects from LULC and reservoir operations, I only see detailed discussion of consumptive use (demand) effects. Demand changes may be associated with LULC change and reservoir operations, but these are fundamentally different things. By the same token, it's great that the authors used monthly streamflow simulations to capture seasonal dynamics, but how did the models incorporate reservoir operation to shift water supply from low-demand months to high-demand months, as is routine water management practice?

We agree with Reviewer #1 that we should have elaborated further on this in the original manuscript. In the revised manuscript we provide, therefore, more detail on the human interventions studied (methods) and we discuss how each of them may impact fresh water resources availability given their known physical relations. Subsequently we evaluate how these human interventions (altogether and individually) affect water scarcity and exposure to water scarcity (see also our response to 1.1 and 1.2). To keep the manuscript concise and focused, however, we do not discuss in detail in our methods section how the different models incorporated the specific human interventions in their modelling structure, we refer here to the specific model papers that describe and discuss this topic in detail.

1.4 Along similar lines, discussion of climate variability and change is virtually absent from the paper. I acknowledge that this is not the main topic of the paper, and that by comparing intervention and non-intervention runs, the effects of climate change are at last partly accounted for. But this still seems like a gap in the analysis. For example, the authors should be able to use a comparison between the intervention and non-intervention runs to estimate how the magnitude of impacts from interventions compare to the impacts of climate change over the 40 year simulation period. It would also be useful to see a few words about how the impacts of interventions may alleviate or exacerbate water scarcity vulnerability to climate change (see recent Hydrol Sci J paper by Fleming). My guess is that the pattern of winners and losers created by interventions, as outlined in the submitted manuscript, might be reinforced by the seasonal flow timing changes expected for most parts of the world under climate change.

As Reviewer #1 already mentions in his/her comment, climate change and climate variability were indeed not the main focus of this paper. In this study, we investigated the historical impacts of human interventions on freshwater resources and water scarcity, given the context of historical developments in socioeconomic and hydro-climatic conditions. Although we think that a 40 year simulation period is somewhat short to study the impacts of climate change (climate is often referred to as the hydro-climatic conditions over a period of 30 or

40 years) we do agree with Reviewer #1 that, in light of this study, the proposed analysis might be a valuable add-on. To accommodate this, we compared the impacts of human interventions on fresh water resources and water scarcity conditions, with the impacts of climate change and climate variability over the same period of time. In the results section of the revised manuscript we have devoted a new paragraph to discuss the outcomes of this analysis. We also discuss how these two drivers of change (human interventions/climate change) may possibly exacerbate/alleviate each other and vulnerability to water scarcity, see the discussion section of our revised manuscript.

1.5 On a more technical note, the instream flow requirement definition used by the authors doesn't appear to widely employed (I've never seen it before, and it looks like only one reference is cited), yet it forms an important part of their water scarcity index. IFN is a large multi-disciplinary field encompassing multiple approaches. In the supplementary materials, I'd like to see a little more explanation provided for the means used to estimate environmental flow needs. This is a question that seems sure to arise for readers who are specialists in areas like hydroecology, habitat assessment, aquatic species diversity, and so forth.

Many thanks for this remark. In our revised manuscript (Supplementary Information) we clarified the method used for EFR calculation and provided more background information with respect to the use of EFR in global water scarcity assessments, its application, and limitations compared to locally derived environmental flow requirements, using the following text:

Instream or environmental flow requirements (EFR, Tessman, 1980; Tennant, 1976; Smakthin et al., 2004) are a relatively young topic in the discipline of global modelling of fresh water resources and water scarcity, with Smakthin *et al.* (2004, 2008) being among the first to include them in global-scale analyses of fresh water resources and basin closure. Smakthin et al. (2004) presented a first attempt to estimate the volume of water required for the maintenance of freshwater-dependent ecosystems at the global scale, using a "combination of ecologically relevant low-flow and high-flow components related to river flow variability and estimated by conceptual rules from discharge time series simulated by the global hydrology model". Building further on their experience and results, various global studies (e.g. Mekonnen & Hoekstra, 2016; Wada et al. 2013; Bonsch et al. 2015) have recently used either a predefined reservation of water (20-70%) or a low-flow indicator (Q90) to take into account a reservation of fresh water for environmental purposes in their fresh water resources and/or water scarcity assessments. Integrating the concept of EFR further in the global modelling of fresh water resources, Pastor et al. (2014) compared and tested different calculation methods for the estimation of EFR in global-scale simulations. In this analysis, the authors compared for 11 case studies the locally defined EFR (with methods being used ranging from: hydrological methods, hydraulic methods, habitat simulation methods; or holistic methods – e.g. expert knowledge) with EFR estimates using a global methodological approach (Tennant, 1976; Smakhtin, 2004; and Tessmann method, 1980). On top of the existing three global methods, two new methods were defined by Pastor et al. (2014) that are combinations of the global methods mentioned before: a purely non-parametric method, using the flow quantiles Q90 and Q50 to estimate EFR, and a parametric method (the variable monthly flow - VMF), following the natural variability of river discharge (following Tessmann, 1980), but increasing the protection of fresh water ecosystems during the low-flow season. According to their validation exercise, the VMF method is one of the best global approaches describing minimum environmental flow requirements at local and regional scales. Given its performance, flexibility, and applicability, we therefore adopted this method in our study. We are, nevertheless, aware of the fact that the environmental flow requirements included in our analysis are relatively simple "conceptual hydrology-based rules-of-thumb for the assessment of bulk environmental water requirements in world river basins" (Smakthin et al., 2004), and that more detailed environmental flow requirements could be incorporated when looking at the scale of individual rivers or basins.

Overall, then, while this paper holds substantial promise, there are a lot of unanswered questions about both methodology (which goes to experimental reproducibility) and implications (which goes to relevance) that need addressing before the article is ready for publication.

Minor comments:

1.6 There are also a few, more minor, points that the authors might address to improve the readability of their article. For example, explanations of several of the figures would benefit from some further work. Figures 2, 3, and 4 all show similar, but not the same, quantities. More direct statements of the punch lines from each of these figures (possibly just within the figure captions themselves) would be helpful to the reader.

Thank you for this suggestion. We updated the captions of these figures accordingly.

1.7 It would also be nice to add a version of Figure 6 giving the same analysis product, but for the non-intervention run. This would enable the reader to more easily tease out intervention effects from climate effects over the multi-decadal model run.

Thank you for this suggestion, climate change effects are now incorporated explicitly in the analysis, discussing its impact relative to that of human interventions in Figure 5 in the revised manuscript (see also response to 1.4)

1.8 The content of Figure 7 could use some clearer explanation; I think this ties back to around line 159 of the manuscript, where a distinction is drawn between sources of change lying upstream of the sub-basin, within the sub-basin, and locally, but without explicitly saying how sub-basins are defined and what exactly is the difference between “within the sub-basin” and “locally.” The reader can make an educated guess, of course, but the ambiguity shouldn’t be there.

Many thanks for this useful comment. Pending the other revisions we omitted this figure out in the revised manuscript. Instead, we now show the dominant source (local or upstream) and driver (which type of human interventions) of significant increases and decreases in fresh water availability globally and regionally, see Figure 3 and Figure 4 in the revised manuscript. We tried, moreover, to clarify better the rationale behind these figures in the text.

1.9 The terminology could also be improved. For example, I strongly suggest using a word other than “persistence” to describe the duration of water scarcity events as defined in this paper. Persistence has a very specific meaning in hydrology and climatology, referring to statistical descriptors of system memory (serial correlation, AR processes, Box-Jenkins models, Hurst effect, etc). I suspect there’s a tie between this meaning and what the authors mean by it, but they’re not the same. Perhaps “event duration” would be better.

Indeed, we agree that misunderstanding might arise from using the word persistence, given its general meaning in the field of hydrology/climatology. In this study, we applied “persistence” as the average length of a water scarcity event. We agree with reviewer #1 that “average duration of water scarcity events” or “event duration” might be more appropriate and we have replaced any reference to persistence throughout the revised manuscript accordingly.

1.10 Similarly, there is a small but unnecessary inconsistency between the definitions for “blue water” provided in the main article versus that in the first paragraph of the supplementary material. Groundwater is included as blue water in the former, but as green water in the latter. Further, this blue vs. green water distinction, though somewhat popular in global water resource studies (particularly in policy papers by social scientists, where perhaps it has some legitimate descriptive use), is fundamentally dubious at an Earth science

level and rarely if ever appears in high-quality local or regional scale hydrologic modeling and water resource planning work. Soil moisture and groundwater (so-called green water) are major physical controls of river flows and lake levels (so-called blue water), and are therefore directly represented (in various fashions) in watershed models simulating surface water resources. Separating the two makes little scientific sense in a modeling study and does not appear to serve a useful function in this article.

Many thanks for this comments. Although we fully agree with reviewer #1 that all fresh water is related (i.e. via the hydrological cycle) and that, from a physical point of view, a distinction between “blue” and “green” water is artificial, we do prefer to keep the term blue water in our manuscript. The term blue and green water are, although perhaps not commonly used in hydrology, essential in global water scarcity studies (see e.g. Kummu et al., 2014; Schyns et al., 2015; Mekonnen & Hoekstra, 2016) and are highly informative to the non-expert reader to indicate the type of water scarcity studied. Although attempts have been made to integrate blue and green water resources in a single water scarcity assessment, only few studies succeeded so far (Kummu et al., 2014) and often blue and green water scarcity are studied in isolation. In our study, we do not evaluate scarcity conditions over rainfed agricultural areas (i.e. green water scarcity) but only take into account water scarcity that arises from shortages in the “blue water” system (i.e. rivers). To avoid any misunderstanding here, we prefer to stick to the term blue water scarcity and/or blue water resources.

Given the inconsistency mentioned by Reviewer #1, we updated the reference to blue water resources in the supplementary information, now including: fresh water resources in “rivers”. For clarity, soil moisture is associated with green water; whilst fossil ground water is often defined as deep blue water (Savenije, 2000). These latter two sources of fresh water (green and deep blue) are not included explicitly in our study.

Reviewer #2:

The paper is well written and very interesting for the reader/community who are particularly working on water scarcity topic. It analysis the impact of human interventions on water scarcity over the period of 1971-2010. The research done for the paper has several novel aspects: I particularly appreciated inclusion of environmental flow requirements and non-human intervention comparisons. The method followed were well described and based on solid scientific background. There are some parts which needs improvement and/or clarification, which are indicated below. If the authors can clarify/change them, the paper will have novel aspects for publication.

We would like to thank reviewer #2 for the encouraging words on our research, and we are pleased that he/she finds the work significant and timely. Below, we address each of the reviewer's issues point-by-point.

Comments:

2.1 Do the authors consider water withdrawal or water consumption for the upstream human intervention? In the abstract, it says water withdrawal but in later stages they mentioned water consumption. It is confusing. This is fundamentally important on water scarcity analysis as upstream water withdrawal may not necessarily affect water availability at the downstream (e.g. return flow and discharge).

In fact, we did include both water consumption and water withdrawals in our analysis. In line with Reviewer #2's rationale, we indeed included upstream water consumption as "human intervention" to diminish fresh water resources. This (i.e. water consumption) is the amount of water that is actually being consumed and that is not returned to the hydrological system (Wada et al., 2011, 2014, 2016). Subsequently, we used water withdrawals to assess water scarcity conditions, since it is the ability to withdraw water from the fresh water systems that determines whether an area experiences water scarcity or not (Raskin et al., 1997; Wada et al., 2011).

2.2 It would be better to explain at the discussion part what does really water scarcity mean. As it is calculated, it actually means reduction of water that is required for environment, does not mean that population experiences any shortages or scarcity. Otherwise exposure to population to water scarcity might be understood differently by readers. Also I questioned if it makes sense to do analysis of "population exposed to water scarcity" since $WS > 1$ can mean either population and/or environment is exposed to WS: an explanation is required to justify this analysis.

Water scarcity is defined in our manuscript as the inability to meet water demands (being either anthropogenic and/or environmental) due to the temporal deficit in fresh water resources (Raskin et al., 1997). In any area of interest, there are normally multiple types of water demands (i.e. domestic water demand, industrial water demand, agricultural water demand, environmental water demand) that compete for the available fresh water resources. If all water demands (from both anthropogenic as well as environmental origin) can be met given the available resources, no water scarcity exists. If too little fresh water is available to meet all the demands (here $WSI > 1$) in a certain region, water scarcity exists and actions should be taken in order to cope.

We agree with the reviewer #2 that, normally, the minimum environmental water requirements are violated first; nevertheless, given the increased attention for the environmental conditions (see e.g. the Sustainable Development Goals), ample examples exist nowadays where anthropogenic water use is being restricted when an area is in water scarcity (e.g. <https://www.sandiego.gov/water/conservation/drought/prohibitions>). Ideally, a more economically based number (e.g. changed welfare levels, economic damages due to water shortages, diminished ecosystem services due to environmental degradation) would be available to characterize the

impact/exposure of water scarcity events globally. In order to do so, one would need, however, to run a full economic-trade-welfare-model, which comes with its own shortcomings (often only driven by a shock (one event), a coarse temporal (decadal time-steps) and spatial resolution (trade regions), highly abstract definition of demand, supply, trade, and allocation of trade networks, and a very limited incorporation of the hydrological system). Applying such a thorough economic-welfare-model for analysis, was –in our opinion- out of the scope of this study, i.e. evaluating the impact of human interventions on the hydrological system (fresh water resources), the critical dimensions of water scarcity events (occurrence, average duration, severity), and the reshuffling of water scarcity hotspots, and we therefore decided to provide insight in the exposure to water scarcity by means of the exposed population only. Nevertheless, we do encourage future research to take the economic impacts into account and we provide recommendations for further study in the discussion section of our revised manuscript.

2.3 I found the main conclusion written by the authors - “adaptation measures should be embedded in integrated river basin management plans, addressing upstream effects on downstream water scarcity” - weak. I mean this is well known by water resources management (both scientists and practitioners) and the authors’ analysis is not needed to draw this conclusion. It is better if the authors include conclusion that uniquely taken out from their study and revise this policy conclusion statement in this current form.

Thank you for this suggestion. Given Reviewer #2’s recommendation we updated our conclusions, providing more detailed recommendations that result from our analysis. We, nevertheless, do find it important to highlight once more that an integrated river basin management plan is key for the successful implementation of adaptation strategies, coping both upstream and downstream with the increased pressure on fresh water resources and their associated consequences.

Reviewer #3

The goal of this study is to evaluate the impacts of human interventions on water resources and scarcity. This is an important research question and the authors have made excellent headway in terms of linking the hydrology models to address this research question. However, when understanding the implications of human activities, it is necessary to model relevant human processes explicitly, rather than inputting human actions as simple geographical grids and proxying for human demands with crude correlations. This is the main reason that I do not find this paper suitable for publication at this time. Additionally, climate information is used to drive pretty much all aspects of this study. Climate is an input variable to the HYDE 3.1/MIRCA datasets used by this study, as well as the socio-economic proxies and hydrology models. This provides climate with a far too prominent role in the methodological framework, attributing most human “interventions” to climate drivers. Instead, human activities that are distinct to climate (not all human actions are driven by climate, but by economics, rational/irrational decision-processes, policies, etc.), but impactful to water resources, should be considered and explicitly modeled.

We thank the reviewer for his/her thorough and fruitful comments. We have made our best efforts to address the very useful recommendations throughout the manuscript. Below, we address each of the major and minor comments point-by-point.

First of all, we would like to stress that we assessed in our study the historical impacts of human interventions on fresh water resources, the critical dimension of water scarcity (occurrence, average duration of events, severity), and its impacts on water scarcity hotspots. The analysis was therefore performed in the light of the historical socioeconomic developments and changes in hydro-climatic conditions throughout the same period of time (1971-2010) and is not focusing on hydro-economic optimization. In doing so, we used a set of 5 state-of-the-art physically-based hydrological models. Given their characteristics, such hydrological models are not (yet) suited for (hydro-)economic optimization or the evaluation of rational/irrational decision processes with respect to water management, and their feedbacks on hydrology, and vice versa. At this moment, the models used do include policy change which is reflected in land use change, population growth, and food production, but these are dominantly one-way feedbacks and not online coupled yet. We underpin the importance of including such analysis in research, but although we see our research field slowly moving towards enabling such analysis, it is too early yet to include such analysis in our study.

To accommodate Reviewer #3's comments related to the attribution of most human interventions to climate drivers, we have included in the revised manuscript an extra analysis in which we compare the impacts of human interventions with the impacts of climate change and hydro-climatic variability over the same period of time. See Figure 5 and the results and discussion section in the revised manuscript.

Major comments:

3.1 Human processes are not explicitly modeled. The physical variables are modeled very effectively. A state of the art approach was used, which is an ensemble of 15 model combinations (5 global hydrology models forced by 3 climate sets) (line 216). However, human processes are not modeled at any point in this paper. Instead, a set of socio-economic proxies is used to estimate human water demand. In fact, “to estimate irrigation, livestock, and industrial water demand, hydroclimatological parameters are used as well” (SI, line 96). So, this modeling framework solely relates demand to socio-economic proxies (which need to be explained better in the SI since these are foundational to the paper) and hydro-climatological parameters. Concepts of supply and demand are foundational in economics. However, this study of water supply and demand does not take economic principles into account. Supply and demand should be impacted by price, not just climate variables. I understand that this is common practice in the water resources and hydrology literature. But, given that the main goal of this paper is to explore inter-linkages between human actions and water scarcity, I find it

necessary to move the methodology forward and explicitly incorporate these key economic concepts, or at least some explicit social processes to the modeling framework.

Thank you for your valuable suggestion. We agree with Reviewer #3 that concepts of supply and demand are foundational in economics and that, ideally, supply and demand should be impacted by price and not just climate and/or hydrological variables. The models used in this study are, however, hydrological models, and are not suited for such cost-minimization or optimization routines, nor for the evaluation of any of these feedback linkages suggested by Reviewer #3. We see our research field slowly moving towards this direction, e.g. by means of hydro-economic or agent-based models – connecting the field of hydrology with economic optimization; trade; social processes; or agent-based decision making problems. Nevertheless, these developments are currently not far enough to include such methodological approaches in a global study like the one executed here.

Moreover, although some economic models (e.g. macro-economic welfare models) might already be able to include the economic processes of demand and supply generation at the global scale, they have their own shortcomings that would limit our analysis (as currently executed) severely. They are relatively crude in terms of spatial (trade regions, countries) and temporal resolution (often decadal time-steps); they often work with predefined rules for economic growth, technological change (similar to what was used in our GHMs), and use as well proxies or econometrically-derived formulas to estimate demand; they often represent trade networks and market operations in an unrealistic way (often via one lump-sum world-market to which all products are exported and from which all products are imported, bilateral trade included in only minor cases). Furthermore, global demands (e.g. for food) are often considered to be fixed (i.e. elasticities in demand not taken into account) and spatial shifts in production and trade should accommodate these global total demands in times of a shock (e.g. water scarcity events in a specific region). Most importantly, such economic models only have a limited representation or ability to represent the hydrological parameters (our main topic of interest) in a realistic manner; are often dependent on decadal shocks to evaluate the societal impacts of hydrological extremes; and only provide a limited representation of the impact of socioeconomic developments on hydrology.

We would like to stress once more that our study specifically focuses on the effects of human interventions on water resources availability and water scarcity conditions, within the context of the historical socioeconomic developments and changes in hydro-climatic conditions. Therefore, we explicitly did not include any feedback linkages generating new water demands. We feel that delving further into such feedback loops and optimization/cost-minimization problems does not fall within the scope of our study (it is a complete research topic itself). We fully agree with Reviewer #3 that for a full hydro-socioeconomic analysis it is important to include all of these aspects and we see opportunities for this in future research. In our revised manuscript, we therefore make a number of suggestions to accommodate the last steps towards such analysis.

Finally, we agree with Reviewer #3 that we should have mentioned more explicitly in the original manuscript how historical water demands (both withdrawal and consumption) were modelled by the different models and that this is not simply done by the use of hydro-climatological and/or socioeconomic proxies. Most (4 out of 5) models studied here have their own water demand model or modelling framework to estimate historical and future water demands using both socioeconomic and hydro-climatological input data. Whilst LPJmL only calculates irrigation water demands.

Livestock water demands are estimated by multiplying the historical gridded livestock counts with their species-specific water daily demands (Flörke et al., 2013; Wada et al., 2014,2016). Domestic water demands were derived using a time-series regression by individual countries and regions using the drivers population and GDP per capita (Flörke et al., 2013; Hanasaki et al., 2008a,b; Pokhrel et al., 2012,2015; Takata et al., 2003;

Wada et al., 2014,2016; Yoshiwaka et al., 2014). PCR-GLOBWB additionally considers total electricity production, energy consumption, and temperature (Wada et al., 2014, 2016). National numbers of domestic water use are distributed to a 0.5° by 0.5° grid according to the gridded total population numbers for all models and technological change rates were considered. Industrial water demands represent water being used for electricity production and/or manufacturing (Flörke et al., 2013; Hanasaki et al., 2008a,b; Pokhrel et al., 2012, 2015; Takata et al., 2003; Wada et al., 2014,2016; Yoshiwaka et al., 2014). Whereas H08 and PCR-GLOBWB based their estimates of industrial water demands on historical country-scale aggregates (electricity production and manufacturing combined) from the WWDR-II (Shiklomanov, 1997; WRI, 1998; Vörösmarty et al., 2005) dataset and the FAO-AQUASTAT (FAO, 2017) database respectively (Hanasaki et al., 2008a,b; Wada et al., 2014,2016; Yoshikawa et al., 2014), WaterGAP simulates global thermoelectric water use using gridded power plant data whilst manufacturing water demand was simulated for each country using the GVA per country and year, a technological change factor, and a manufacturing structural water use intensity (Flörke et al. 2013). Irrigation water use, finally, was estimated generally multiplying the area equipped for irrigation with the utilization intensity of irrigated land, the total crop water requirements per unit of irrigated area and the efficiency of irrigation that accounts for the losses during water transport and application of the irrigation method (Wada et al., 2016). Here, the specific crop water requirements are driven by the hydro-climatic conditions (temperature, precipitation, potential evapotranspiration, soil moisture, crop-growth curves, length and timing of the crop-growth season), whilst irrigation efficiency, the area equipped for irrigation and the utilization intensity are merely determined by economic, technological and political factors (Bondeau et al., 2007; Döll et al., 2014; Döll & Müller Schmied, 2012; Döll & Siebert, 2002; Hanasaki et al., 2008a,b; Müller Schmied et al., 2014, 2016; Pokhrel et al., 2012, 2015; Portmann et al., 2010; Schaphoff et al., 2013; Takata et al., 2003; Wada et al., 2014). Each of these global hydrological models have, moreover, performed an extensive validation study on the ability of their water demand model/modelling frameworks to reflect historical developments in water demands, using historical observations or reported values as validation datasets, see Wada et al. (2014, 2016), Hanasaki et al. (2008a,b), Flörke et al. (2013), Pokhrel et al. (2012).

In the Supplementary Methods and Data we describe (like above) how the historical water demands used in our study were modelled and discuss that they have been validated extensively in previous research. To keep the manuscript itself focused and concise we use within the main body of text predominantly references to the papers describing the individual models in detail.

3.2 Irrigation input is fixed in time. The paper seeks to evaluate human impacts over the period 1971-2010. However, the MIRCA2000 irrigation data set -- which is fixed to the time period around the year 2000 -- was used to force the hydrology models. This is problematic. Additionally, the HYDE 3.1 database was used (line 226) but it is not clear if an annual grid was obtained from HYDE 3.1 or some other temporal average was used for this geographical input. According to the SI, line 101: "All impact models modelled irrigation water consumption and withdrawals endogenously, using the dynamic HYDE. 3.1/MIRCA dataset as base-map for the extent of irrigated areas." However, the references provided for these databases on line 226 are references 38, 39, and 40. From reference 39 the MIRCA2000 database is fixed in time. So, it is not clear why these datasets are called "dynamic". This needs to be more clearly explained in future revisions.

We agree with Reviewer #3 that this was not completely clear from the original manuscript. What was used in this study is a combination of present-day (year 2000) rainfed and irrigated areas from MIRCA 2000 (Portmann et al., 2010) cropland and pasture extents from Ramankutty et al. (2008), and trends of agricultural land from HYDE 3 (Klein Goldewijk & Van Dreht, 2006). The dynamic, or time-varying, dataset on cropland area and irrigated areas was eventually produced following the method of Fader et al. (2010). Therein, historical patterns of irrigated and rainfed crop areas were identified for the following crop and vegetation classes: temperate cereals, rice, maize, tropical cereals, pulses, temperate roots, oil crops – sunflower, oil crops – soybean, oilcrops – groundnut, oilcrops – rapeseed, sugarcane, pastures, others. These gridded historical

patterns of irrigated and rainfed crop-specific agricultural areas were subsequently used as input in the global hydrological models to model -on the one hand-agricultural water demands, and – on the other hand – impacts on fresh water resources (due to e.g. changes in infiltration/storage capacity, seepage to sub-surface layers, runoff generation, and evapotranspiration rates). In the revised manuscript we updated our description of this dataset, accordingly.

3.3 Groundwater interactions are not clear. The study claims to evaluate water availability from groundwater sources (line 230). However, none of the results of the paper highlight groundwater resources, which should dampen water scarcity outcomes. All results are given in terms of surface watersheds. The ability to access groundwater resources would enable populations to minimize their vulnerability to local water scarcity in both space and time. It is not clear if this study enables groundwater pumping, and, if so, how and when groundwater pumping interacts with water scarcity outcomes. One of the main conclusions of the study is that upstream water withdrawals lead to significant downstream reductions in water availability (line 132). It is important to understand if these downstream pixels are able to adapt and counteract this by pumping groundwater.

We agree with Reviewer #3 that we did not deal with the groundwater interactions explicitly enough in the original manuscript. In the revised manuscript we clarified this accordingly. The “groundwater sources” that were taken into account indirectly in this study are aquifers that feed the generation of runoff and discharge via lateral flow (sub-surface runoff). Those not taken into account as a freshwater source directly on the supply side are the deep ground water resources that do not feed the runoff and discharge estimates. The reservoirs referred to in line 230 are human-made and human-managed fresh water surface-reservoirs. All hydrological models applied in this study are bucket-type water balance models that include between 1 (H08) and 13 soil layers (MATSIRO), representing both the soil and the shallow aquifers. Effective precipitation ($P - ET$) passes these soil layers, and part of it forms the input for the volume of sub-surface/lateral runoff through the soil and shallow aquifers and eventually feeds the runoff and discharge volumes used in this study. Another fraction ends up in the deep groundwater reservoirs via seepage – the size of these reservoirs is often not specified by the individual hydrological models. Within this portion of deep groundwater we can distinguish, however, a fraction of non-renewable groundwater (often not specified by the hydrological models) and a fraction of renewable groundwater (equal to the recharge levels).

On the water demand side, we do account for the historical non-renewable groundwater abstractions, however, following the rationale of Wada et al. (2011): “The amount of groundwater that is abstracted in excess of recharge will, albeit temporally and non-renewably, decrease the demand for blue water and mitigates water stress”. To estimate the fraction of non-renewable ground water abstractions taking place historically, Wada et al. (2011) used downscaled country-based data on groundwater abstractions while accounting for irrigation recharge. In this study, we used these resulting spatially- and temporally-explicit non-renewable groundwater abstraction rates to diminish the anthropogenic demands for fresh water.

As previously mentioned, this study provides a historical analysis on the impacts of historical human interventions on fresh water resources and water scarcity and therefore we think it is out of the scope of this study to explicitly evaluate whether downstream areas under water scarcity are or are not able to minimize the adverse impacts of human interventions by increasing or expanding local ground water abstractions.

Minor comments:

3.4 According to the SI, line 106: “Environmental flow requirements were reduced according to the actual irrigation water consumption, while WSI was calculated using the potential irrigation water withdrawals”. This seems inconsistent.

Given the Reviewer #3's comment on inconsistency we agree that we should clarify this in the revised manuscript. The difference between actual irrigation water consumption and potential irrigation water withdrawals is comparable to the difference between water withdrawals and the actual water consumption and can best be explained with a numeric example. Whilst 100 liter of water is being withdrawn from the river system initially, only 80 liter is actually consumed (by industry, for domestic purposes, or here: for irrigation). The remaining 20 liter flows –more or less directly – back into the hydrological system, either into the river system (industry, domestic) or into the soil layers (irrigation). For irrigation, we used therefore the actual irrigation water consumption to assess the eventual impact of irrigation activities on the hydrological system. To evaluate water scarcity, however, we use the potential irrigation water withdrawals, since this is the volume of water that should be initially available to meet the irrigation water demands.

3.5 Exposure of humans to water scarcity. This is done by co-locating human population pixels with quantified values of water scarcity across all economic sectors. It seems more reasonable to quantify populations that live in locations that have high water stress due to only domestic demands outstripping local water availability.

Thank you for this suggestion. Following the definition of water scarcity, scarcity occurs if the sum of both anthropogenic (domestic, industry, and agriculture) and environmental water demands outweigh the available fresh water resources ($WSI > 1$). Although we understand Reviewer #3's suggestion that it might be more appropriate to relate human exposure only to water scarcity caused by domestic water use, this does not seem to fit the formal definition of water scarcity. Domestic water use is only a small water demanding sector and is hardly ever the only determinant of water scarcity. Societies experiencing water scarcity do so because the sum of all competing water demands (including the environmental needs) is too large given the available fresh water resources. Society is just as affected by water scarcity induced primarily by e.g. irrigation activities, as by industrial activities or relatively high environmental requirements that put pressure on the available fresh water resources. Although we agree with Reviewer #3 that, ideally, water scarcity should be expressed in more economic terms (see also our response to 3.1) in which we could distinguish, for example, the economic impacts of an industrial based water scarcity event from that of a domestic one, methodologies to do so at a global scale are, unfortunately, not fully developed yet and not available for use in this analysis. An often used alternative to describe the societal exposure to water scarcity is to express water scarcity by means of the population being exposed to water scarcity events (Mekonnen & Hoekstra, 2016; Wada et al., 2011), an approach that was used in this study.

References used in this response:

- Bondeau, A. *et al.* Modelling the role of agriculture for the 20th century global terrestrial carbon balance. *Global Change Biology* **13**, 679–706 (2007).
- Bonsch, M. *et al.* Environmental flow provision: Implications for agricultural water and land-use at the global scale. *Glob. Env. Change* **30**, 113-132 (2015).
- Döll, P. & Siebert, S. Global modelling of irrigation water requirements. *Water Resour. Res.* **38**, W1037 (2002).
- Döll, P. & Müller Schmied, H. How is the impact of climate change on river flow regimes related to the impact on mean annual runoff? A global-scale analysis. *Environ. Res. Lett.* **7**, 014037 (2012).
- Döll, P., Fritsche, M., Eicker, A. & Müller Schmied, H. Seasonal Water Storage Variations as Impacted by Water Abstractions: Comparing the Output of a Global Hydrological Model with GRACE and GPS Observations. *Surv. Geophys.* **35**, 1311-1331 (2014).
- Fader, M., Rost, S., Müller, C., Bondeau, A. & Gerten, D. Virtual water content of temperate cereals and maize: Present and potential future patterns. *Journal of Hydrology* **384**, 218-231 (2010).
- FAO – Food and Agriculture Organization of the United Nations. AQUASTAT online database, available at: <http://www.fao.org/nr/water/aquastat/dbase/index.stm>, last access: 16th January 2017.
- Flörke, M. *et al.* Domestic and industrial water uses of the past 60 years as a mirror of socio-economic development: A global simulation study. *Global Environ. Change* **23**, 144–156 (2013).
- Hanasaki, N. *et al.* An integrated model for the assessment of global water resources – Part 1: Model description and input meteorological forcing. *Hydrol. Earth Syst. Sci.* **12**, 1007–1025 (2008a).
- Hanasaki, N. *et al.* An integrated model for the assessment of global water resources – Part 2: Applications and assessments. *Hydrol. Earth Syst. Sci.* **12**, 1027–1037 (2008b).
- Klein Goldewijk, K. & Van Drecht, G. HYDE 3: Current and historical population and land cover. Integrated modelling of global environmental change. An overview of IMAGE 2.4 (Eds: A.F. Bouwman, T. Kram and K. Klein Goldewijk). Netherlands Environmental Assessment Agency (MNP), Bilthoven, The Netherlands (2006).
- Kummu, M., Gerten, D., Heinke, J., Konzmann, M. & Varis, O. Climate-driven interannual variability of water scarcity in food production potential: a global analysis. *Hydrol. Earth Syst. Sci.* **18**, 447–461 (2014).
- Mekonnen, M.M. & Hoekstra, A.Y. Four billion people facing severe water scarcity. *Sci. Adv.* **2**, e1500323 (2016).
- Müller Schmied, H. *et al.* Sensitivity of simulated global-scale freshwater fluxes and storages to input data, hydrological model structure, human water use and calibration. *Hydrol. Earth. Syst. Sci.* **18**, 3511-3538 (2014).
- Müller Schmied, H. *et al.* Variations of global and continental water balance components as impacted by climate forcing uncertainty and human water use. *Hydrol. Earth Syst. Sci.* **20**, 2877-2898 (2016).
- Pastor, A. V., Ludwig, F., Biemans, H., Hoff, H. & Kabat, P. Accounting for environmental flow requirements in global water assessments. *Hydrol. Earth. Syst. Sci.* **18**, 5041-5059 (2014).

- Pokhrel, Y.N. *et al.* Incorporating anthropogenic water regulation modules into a land surface model. *J Hydrometeor* **13**, 255–269 (2012).
- Pokhrel, Y.N. *et al.* Incorporation of groundwater pumping in a global land surface model with the representation of human impacts, *Water Resour. Res.* **51**, 78–96 (2015).
- Portmann, F.T., Siebert, S. & Döll, P. MIRCA2000 – Global monthly irrigated and rainfed crop areas around the year 2000: A new high-resolution data set for agricultural and hydrological modelling. *Global Biogeochem. Cycles* **24**, GB1011 (2010).
- Ramankutty, N., Evan, A.T., Monfreda, C. & Foley, J.A. Farming the planet: 1. Geographic distribution of global agricultural lands in the year 2000. *Global Biogeochem. Cycles* **22**, GB1003 (2008).
- Raskin, P., Gleick, P., Kirshen, P., Pontius, G. & Strzepek, K. Comprehensive assessment of the freshwater resources of the world. Stockholm Environment Institute, Stockholm, Sweden (1997).
- Savenije, H.H.G. Water scarcity indicators; the deception of the numbers. *Phys. Chem. Earth (B)* **25**, 199-204 (2000).
- Schaphoff, S. *et al.* Contribution of permafrost soils to the global carbon budget. *Env. Res. Lett.* **8**, 014026 (2013).
- Schyns, J.F., Hoekstra, A.Y. & Booij, M.J. Review and classification of indicators of green water availability and scarcity. *Hydrol. Earth. Syst. Sci.* **19**, 4581-4608 (2015).
- Shiklomanov, I.A. Assessment of water resources and water availability in the world, Comprehensive assessment of the freshwater resources of the world. World Meteorological Organization and the Stockholm Environment Institute (1997).
- Smakhtin, V., Revenga, C. & Döll, P. A pilot global assessment of environmental water requirements and scarcity. *Water Int.* **29**, 307–317 (2004).
- Smakhin, V. Basin closure and environmental flow requirements. *Internat. J. of Wat. Res. Dev.* **24**, 227-233 (2008).
- Takata, K., Emori, S. & Watanabe, T. Development of minimal advanced treatments of surface interaction and runoff. *Global Planet. Change* **38**, 209–222 (2003).
- Tennant, D. L. Instream flow regimens for fish, wildlife, recreation and related environmental resources. *Fisheries* **1**, 6–10 (1976).
- Tessmann, S. Environmental assessment, technical appendix e in environmental use sector reconnaissance elements of the western dakotas region of south dakota study. South dakota state university, Water Resources Institute, South Dakota State University, Brookings, South Dakota (1980).
- Vörösmarty, C. J., Leveque, C. & Revenga, C. Millennium Ecosystem Assessment Volume 1: Conditions and Trends, chap. 7: Freshwater ecosystems. Island Press, Washington DC, USA, 165–207 (2005).
- Wada, Y., Van Beek, L.P.H. & Bierkens, M.F.P. Modelling global water stress of the recent past: on the relative importance of trends in water demand and climate variability. *Hydrol. and Earth Syst. Sci.* **15**, 3785–3808 (2011).
- Wada, Y., Van Beek, L.P.H., Wanders, N. & Bierkens, M.F.P. Human water consumption intensifies hydrological drought worldwide. *Env. Res. Lett.* **8**, 034036 (2013).

Wada, Y., Wisser, D. & Bierkens, M.F.P. Global modelling of withdrawal, allocation and consumptive use of surface water and groundwater resources. *Earth Syst. Dyn.* **5**, 15–40 (2014).

Wada, Y. *et al.* Modeling global water use for the 21st century: Water Futures and Solutions (WFaS) initiative and its approaches. *Geosci. Model Dev.* **9**, 175-222 (2016).

World Resources Institute. World Resources: A Guide to the Global Environment 1998–99. World Resources Institute, Washington DC, USA (1998).

Yoshikawa, S., Cho, J., Yamada, H.G., Hansaki, N. & Kanae, S. An assessment of global net irrigation water requirements from various water supply sources to sustain irrigation: rivers and reservoirs (1960-2050). *Hydrol. Earth Syst. Sci.* **18**, 4289-4310 (2014).

Reviewers' Comments:

Reviewer #2:

Remarks to the Author:

I think authors improved the manuscript significantly in terms of clarity, description of methods used, modelling approach etc. and addressed most of the comments that were provided by the reviewers. I would like to thank the authors for this major revision. I also highly appreciate the work done, particularly improved methodology, new novelty items in modelling and analytical framework. However, I still struggle about novelty of the key findings of this study. When I read the manuscript, I can clearly see the improvements introduced in methodology, datasets, modelling and analytical work compared to existing studies/literature, but I have difficulties in seeing what the article reveals that is not known previously (except quantification part). Human interventions focusing on supply affect water scarcity positively (decreasing it) and HIs related to demand (mostly) affect water scarcity negatively. Logically, in some places measures taken for supply overcome the pressure put by the demand and in some places demand is too much to handle only by increase in supply. Of course, downstream population is affected negatively compared to the upstream ones. We all know this. I am not fully convinced why quantifying this knowledge is important to know as presented in the study. I think the article in this current form still lacks of novelty aspect in terms of conclusion/messaging.

Reviewer #3:

Remarks to the Author:

The authors have adequately addressed my concerns. I find the paper to now be suitable for publication.

Reviewer #2 (Remarks to the Author):

I think authors improved the manuscript significantly in terms of clarity, description of methods used, modelling approach etc. and addressed most of the comments that were provided by the reviewers. I would like to thank the authors for this major revision. I also highly appreciate the work done, particularly improved methodology, new novelty items in modelling and analytical framework. However, I still struggle about novelty of the key findings of this study. When I read the manuscript, I can clearly see the improvements introduced in methodology, datasets, modelling and analytical work compared to existing studies/literature, but I have difficulties in seeing what the article reveals that is not known previously (except quantification part). Human interventions focusing on supply affect water scarcity positively (decreasing it) and HIs related to demand (mostly) affect water scarcity negatively. Logically, in some places measures taken for supply overcome the pressure put by the demand and in some places demand is too much to handle only by increase in supply. Of course, downstream population is affected negatively compared to the upstream ones. We all know this. I am not fully convinced why quantifying this knowledge is important to know as presented in the study. I think the article in this current form still lacks of novelty aspect in terms of conclusion/messaging.

We thank Reviewer #2 for his/her time taken to review the manuscript once more. We are happy to see that he/she finds that the manuscript has greatly improved after this round of revisions. In response to Reviewer #2's last comments we have rephrased our abstract, and parts of the introductions and conclusion, putting more emphasis on the novelty and the added value of our work. We slightly revised the title of our study accordingly. The following points summarize the main novelties/advancements of our study. Each of these points is now explicitly incorporated and discussed in the manuscript:

- 1) In this study we show and quantify for the first time how the critical dimensions of water scarcity change due to human interventions, looking at the exposure to and occurrence and duration of water scarcity events. Whereas previous assessments have studied the relation between human interventions and streamflow or evapotranspiration, we have made the translation to water scarcity and provide information on how crucial water scarcity characteristics change due to human interventions. Insights that can be potentially more informative for policy makers than the hydrological parameters used in previous studies.
- 2) Moreover, we introduced a spatially and temporally explicit measure of the minimum environmental flow requirement. Together with our seasonal assessment of water availability and water scarcity conditions, accounting for seasonal variability and regional variation at a high spatial-resolution, we developed an updated Water Scarcity Index (WSI). This index provides a more meaningful indicator for water scarcity conditions at the seasonal scale than those used in past studies, as it reflects both the human and environmental water needs.
- 3) In this study we show that water scarcity travels downstream due to the unequal impacts of human interventions (HI) between upstream and downstream basins, with alleviated water scarcity for 8.3% of the global population but aggravated conditions elsewhere for 8.8%. Moreover we show that more than one-third of the global population experienced a significant increase in the average duration and occurrence of water scarcity events, respectively. Quantification is important and novel here, in our opinion, as these processes are often overlooked, especially in global studies. Putting a number on both the positive and negative impacts, but also on the uncertainty range, can potentially provide more transparency to policy makers dealing with water scarcity and deciding over implementation of HI (see also 4, 5 and 6).
- 4) In this study we disentangle the dominant drivers and sources of changes in water scarcity due to human interventions. Not only facilitates this a better understanding in the specific dynamics of human interventions on water scarcity and the credibility of water scarcity assessments under transient simulations. It also provides insights into the management options dealing with current and future water scarcity, as well as the basin-wide consequences of such strategies that should be taken into account before implementing them.

- 5) Both the seasonal variation in exposure to water scarcity and the impacts of human interventions, as well as the discrepancy between upstream and downstream effects, dominant driving forces, and sources of change, highlight further that the relation between HI and water scarcity is complex and not univocal and that thorough consideration of such interactions is crucial for successful water scarcity adaptation.
- 6) The multi-model multi-forcing approach used in this study enabled us, finally, to evaluate the robustness of results across the different models and forcing datasets. The impacts of HI are globally significant, both with respect to the changes in water availability, as well as regarding the changes in exposure to water scarcity (movement in/out, aggravation/alleviation), and the changes in average duration and occurrence of water scarcity events. The found modelling-spread is, nevertheless, a call for the different global models to further improve the incorporation of human interventions in their modelling framework, thereby accounting for possibly interaction effects and feedback linkages.

Reviewer #3 (Remarks to the Author):

The authors have adequately addressed my concerns. I find the paper to now be suitable for publication.

We would like to thank Reviewer #3 for taking the time to review again the revised manuscript. We are pleased that he/she shares our feeling that that the manuscript has greatly improved based on the comments and subsequent revisions. We are delighted that he/she find it now suitable for publication.